# Multiscale Entropy as a New Feature for EEG and fNIRS Analysis

**DOI:** 10.3390/e22020189

**Published:** 2020-02-07

**Authors:** Thanate Angsuwatanakul, Jamie O’Reilly, Kajornvut Ounjai, Boonserm Kaewkamnerdpong, Keiji Iramina

**Affiliations:** 1Graduate School of Systems Life Sciences, Kyushu University, Fukuoka 819-0395, Japan; iramina@inf.kyushu-u.ac.jp; 2College of Biomedical Engineering, Rangsit University, Pathum Thani 12000, Thailand; 3Biological Engineering Program, Faculty of Engineering, King Mongkut’s University of Technology Thonburi, Bangkok 10140, Thailand; kajornvut.oun@kmutt.ac.th (K.O.); boonserm.kae@kmutt.ac.th (B.K.); 4Faculty of Information Sciences and Electrical Engineering, Kyushu University, Fukuoka 819-0395, Japan

**Keywords:** brain complexity, electroencephalogram (EEG), functional near-infrared spectroscopy (fNIRS), multiscale entropy (MSE)

## Abstract

The present study aims to apply multiscale entropy (MSE) to analyse brain activity in terms of brain complexity levels and to use simultaneous electroencephalogram and functional near-infrared spectroscopy (EEG/fNIRS) recordings for brain functional analysis. A memory task was selected to demonstrate the potential of this multimodality approach since memory is a highly complex neurocognitive process, and the mechanisms governing selective retention of memories are not fully understood by other approaches. In this study, 15 healthy participants with normal colour vision participated in the visual memory task, which involved the making the executive decision of remembering or forgetting the visual stimuli based on his/her own will. In a continuous stimulus set, 250 indoor/outdoor scenes were presented at random, between periods of fixation on a black background. The participants were instructed to make a binary choice indicating whether they wished to remember or forget the image; both stimulus and response times were stored for analysis. The participants then performed a scene recognition test to confirm whether or not they remembered the images. The results revealed that the participants intentionally memorising a visual scene demonstrate significantly greater brain complexity levels in the prefrontal and frontal lobe than when purposefully forgetting a scene; *p* < 0.05 (two-tailed). This suggests that simultaneous EEG and fNIRS can be used for brain functional analysis, and MSE might be the potential indicator for this multimodality approach.

## 1. Introduction

Each neuroimaging modality has its own peculiarities, pros and cons. For example, electroencephalogram (EEG) recordings give high temporal resolution but low spatial resolution, whereas functional near-infrared spectroscopy (fNIRS) offers high spatial resolution but relatively low temporal resolution (compared with EEG). Furthermore, fNIRS does not suffer from electromyographic (EMG) and blink artefacts. Combining the two modalities of EEG and fNIRS is therefore a complementary approach for brain function analysis wherein the strengths of each method support the weaknesses of the other. The simultaneous acquisition of EEG and fNIRS may therefore provide a more complete picture of brain activity than either of the two individual modalities. In this sense, we believe that fNIRS should not be considered a competitor of EEG, but rather as a pooling modality for cases in which one technique is insufficient.

For an in-depth understanding and insight of what is happening inside the human brain, both high spatial and high temporal resolutions are likely to be beneficial. Research studies combining simultaneous EEG/fNIRS to examine memory functions are scant. However, there are a handful of published studies using simultaneous EEG and fNIRS to assess visual memory function [1,2,3], which employ bimodal recordings of EEG and fNIRS to measure and discriminate between levels of mental workload; promising results for the accurate quantification of mental workload in real-world settings are reported therein. In addition, Sun et al. [4] proposed a multimodal approach to evaluate EEG and fNIRS signals for affective state detection. Their results indicate that simultaneous EEG and fNIRS improves performance over fNIRS or EEG only. Despite using bimodal recordings of EEG and fNIRS, neither of these studies has explored the application of multiscale entropy (MSE) for signal analysis.

### 1.1. EEG as Measurement for Brain Complexity

Electroencephalography is regarded as a noninvasive, functional neuroimaging technique [5], which consists of measuring electrical activity from the brain using electrodes placed on the scalp. This has been widely applied in order to investigate the underlying processes involved in neural activity. Since EEG can measure brain activity with high temporal resolution, it is suitable for capturing dynamic brain activity. Recording EEG remains the most effective technique for observing near-instantaneous neural responses to visual stimuli [6]. 

Furthermore, EEG devices are inexpensive and robust for the measurement of brain activity [7]. However, the fundamental limitation of EEG is the low spatial resolution caused by the diffusion of electrical signals travelling between their intracranial source and the electrodes on the scalp. The inverse square law of electric fields also limits the influence of subcortical structures on EEG waveforms. Generally, theta (4–7 Hz) and alpha (8–12 Hz) frequency ranges are dominant during working memory tasks [8]. Increases in theta range spectral power have been associated with increasing difficulty of memory tasks [9].

In addition, Zhang et al. [10] evaluated the modulation effects of mental workload on functional connectivity and found sustained and load-dependent theta power enhancement in the frontal midline region. In addition, Teng et al. [11] compared EEG recordings recorded during resting and active stages in elderly people, and they found that beta spectral power significantly decreased. This change may be a reflection of age-related changes in memory.

### 1.2. fNIRS as Measurement for Brain Complexity

Near infrared spectroscopy (NIRS) is a relatively recent invention compared with EEG. It was developed in 1992 and presently multichannel systems are commercially available [12]. As a neuroimaging tool, functional NIRS (fNIRS) offers a number of advantages. For example, it is noninvasive, easy to set up and does not require a large space [13]. Moreover, it offers a reasonable trade-off between temporal and spatial resolution [14], improving on the spatial resolution of EEG. This method is also safe for use with neonates [15] and can be applied to observe brain function over extended periods.

To perform fNIRS, two near-infrared lights (650 and 1000 nm) are emitted through the scalp. Photodetectors track the diffusion of NIR light energy through the underlying brain tissue by measuring light intensity. This measurement relies on the absorption of light and light-scattering properties of the tissue. Changes in the metabolic activity of the brain induce haemodynamic responses that alter the rate of NIR light absorption. This influences the level of NIR light intensity being measured, which provides an index of oxy-haemoglobin (Hb.O2) and deoxy-haemoglobin (Hb.R) concentrations [16,17]. These are related to local neural function and the presumed cause of metabolic demand.

Functional near-infrared spectroscopy (fNIRS) was used by Borfeld et al. [18] to study brain activity in infants. They found hemispheric asymmetry in the level of Hb.O2, with the left more active while the infants were watching an animation with speech. Schroeter et al. [19] investigated brain functions during a Stroop test using fNIRS. They found a left-lateral difference in the haemodynamic responses of the prefrontal cortex that emerged with age. 

Araki et al. [20] used fNIRS to analyse brain function in children diagnosed with hyperactivity disorder in response to treatment with atomoxetine. Levels of Hb.O2 showed a significant decrease in the left ventrolateral prefrontal cortex. This neuroimaging modality has also been applied to investigate changes in brain activity following rehabilitation therapy in stroke patients.

According to Schreppel et al. [21], fNIRS has been widely used to investigate neural responses. They identified changes in several brain regions, including the prefrontal cortex, occipital region, parietal regions and medial temporal areas. Using fNIRS, it has been found that differences in brain activity of the frontal cortex depend on the level of difficulty of tasks [22]. Additionally, oxy-haemoglobin concentration during difficult tasks changes more when compared with easy tasks. This has been confirmed through investigation; for example, a study by Sasabe and Hagiwara [23] characterised and compared oxygenated haemoglobin concentration changes in the brain using fNIRS measurements. Their analysis demonstrated a significant correlation between the number of false answers and the level of Hb.O2, suggesting that Hb.O2 increases with task difficulty level. 

Mirelman et al. [24] also found a difficulty-related increase in frontal brain activity while doing two tasks during walking. Increased Hb.O2 in the frontal cortex was maximal while the participants were walking and serially subtracting 7s, followed by walking while counting forward, then the lowest while the participants were walking or standing without performing mental calculations. These kinds of studies rely on well-defined and separable task difficulties; if tasks are ill defined or too similar, the analysis may be obscured.

### 1.3. Potential Application in the Study of Memory

Memory plays a crucial role in human life. It allows us to construct life-narratives by linking past, present and future events. Past events that we have experienced are important because they enable us to make educated decisions in the present and to construct mental scenarios in real-time; this forward thinking, or ability to simulate future events, is a quintessentially human trait [25]. Memory can be classified into long-term, short-term and working memory (WM) [26]. Long-term memory, in which information is held indefinitely, encrypts information semantically for storage. In contrast, short-term memory stores information temporarily for fast response activities. Short-term memory retains information for a short period, whereas WM manipulates this for temporary storage. In other words, WM retains and uses, while short-term merely retains. This has been examined in previous functional neuroimaging studies, which identified the prefrontal cortex (PFC) as the most relevant cortical locus of activation [27]. Memory also changes through time; some grow stronger and last, whereas others fade.

Central to the functioning of the mind, memory is associated with abilities and outcomes such as intelligence and educational achievement and is linked to sensory processes [28]. Different studies have been carried out to understand memory functions based on fNIRS and EEG independently, all of which have their own strengths and weaknesses. For example, Aghajani et al. [3] and Aghajani and Omurtag [29] measured mental workload during a working memory task while recording EEG and fNIRS and identified some clear differences between the two signals. They correctly point out that EEG results from the summed electrical potential of large numbers of cortical and some subcortical neurons with submillisecond temporal resolution. In contrast, fNIRS yields localised measures of Hb.O2 and Hb.R concentrations, which are considered to reflect metabolic changes associated with neural activity. Furthermore, the physical measurement of EEG and fNIRS is somewhat different and requires conductive electrodes and light-sensitive transducers, respectively. This distinction in the signal acquisition method causes EEG to be prone to distortion from blink and muscle artefacts. However, it is not the case for fNIRS.

Coffey et al. [30] used three-channel fNIRS covering the left forehead together with eight EEG electrodes placed in the frontal and central areas, in an n-back task. They extracted EEG power spectra and fNIRS Hb amplitude features, and employed them in a linear discriminant analysis classification. The results from ten participants revealed a maximum accuracy of 79.7% (with fNIRS) and 89.6% (with EEG). Moreover, Liu et al. [31] employed a 16-optode continuous wave fNIRS system covering the forehead and 28 EEG electrodes positioned according to the standard 10/20 system in the n-back task. The results from sixteen participants showed significant correlations between WM load and EEG as well as Hb.O2 and Hb.R.

### 1.4. Hypotheses and Objectives

The present study focuses on examining brain activity using simultaneous EEG/fNIRS data recorded during a behavioural task. Multiscale entropy (MSE) was then applied to analyse these signals in terms of entropy or brain complexity levels. Brain complexity is considered as an interesting characteristic since the dynamics of brain activity can illustrate task-related functions. In this study, a pooling of bioelectricity (EEG) and haemodynamic (fNIRS) signals was used to measure different perspectives of brain complexity. The scale of data measurement was also adjusted to be consistent with the entropy scale.

The main aim of the present study was to apply MSE as a new feature for EEG and fNIRS analysis in respect to brain complexity levels. A null hypothesis is formulated as follows: there is no statistically significant difference in the participants’ responses to pictures with visual stimuli between MSE analysis of EEG/fNIRS measurements.

## 2. Materials and Methods

### 2.1. Multiscale Entropy (MSE) 

The notion of entropy was first introduced by Rudolph Clausius in the context of thermodynamics [32]. It acts in accordance with the second law of thermodynamics, which states that the change of the entropy in the volume element is equivalent to the ratio of the heat state changes in temperature [33]. The term “Entropy” is thus appreciated from the viewpoint of both thermodynamics and information theory by means of a display of system uncertainty. The concept of information entropy is proposed by Shannon in 1948, who is regarded as the father of information theory. At present, the applications of entropy in biomedical signals have received extensive research attention [34,35,36]. For instance, Pincus [37] proposed the concept of approximate entropy to examine the changes of heart rate for infantile sudden illness. Approximate entropy is well suited to solve the problem of common signal with short noise in biomedical signals. Richman and Moorman [38] developed sample entropy. Sample entropy, compared with approximate entropy, shows better relative consistency and less dependence on data length [38,39,40].

On the basis of sample entropy with a multiscale approach on temporal data, MSE has been widely applied for medical data analysis. This is pertinent given the complexity and nonlinearity of medical data. Shannon entropy has led to the development of various entropies such as Kolmogorov–Sinai entropy, Grassberger entropy and approximate entropy. Approximate entropy (ApEn) has been commonly used to analyse biological data [41,42]. Furthermore, it was developed into sample entropy (SampEn) [38,42]. In the vein of ApEn, SampEn divides the data into short sequences, called epochs, and then determines the sample entropy according to Equation (1).
(1)$$\mathrm{SampEn}\left(m,r,N\right)=-\log\left(\frac{\sum A_{i}}{\sum B_{i}}\right)=-\log\frac{A}{B},$$
where *m* is template length, *r* is the tolerance for accepting matches, *N* is the number of data points, *A_i_* is the number of matches of length *m+1* and *B_i_* is number of matches of length *m*.

To determine MSE, the data are coarse-grained by averaging in the range of a specified scale; for scale 2, two adjacent data points are averaged, thus the number of total data points is reduced to N/2 if there are originally a total of N data points. Through the scaling process, the new dataset can be expressed by Equation (2) [38,42,43,44,45]. As a result of a large number of recording channels with a high sampling frequency, multiscale analysis is required to obtain linear MSE graphs. Sample entropy was therefore selected to optimise efficiency in data processing [38].
(2)$$y_{j}^{\left(\tau \right)}=\frac{1}{\tau }\sum_{i=j\tau -\tau +1}^{j\tau }x_{i},\;1\leq j\leq \frac{N}{\tau },$$
where *τ* is the scale level and *x_i_* is the original data point, then the new dataset is used to compute sample entropy in each scale.

Previous studies have shown that SampEn has the acceptable range for *m* = 2, *r* = 0.15 × SD of the time series [38,46]. Adopting the method proposed by Catarino et al. [46], we used *m* = 2, *r* = 0.15 × SD of the time series. For *N*, we calculated the sampling frequency and epoch length used for quantitative analysis (12 s). 

According to Richman and Moorman [38], the appropriate *N* size for SampEn analysis must be a minimum of 10*^m^* to 20*^m^*, where *m* = 2; *N* should be at least 100 to 400 in each scale. When the scale increases, the number of *N* in each scale will decrease. For an EEG sampling frequency at 500 Hz, when calculating the size of *N*, *N_EEG_* is equal to 500 Hz × 12 s = 6000 data points. Considering the 20th scale, the size of the *N_EEG_* is 300, which is still enough for SampEn analysis. With respect to fNIRS, which has a sampling frequency of only 10 Hz. To calculate the size of *N*, *N_fNIRS_* = 10 Hz × 12s = 60 data points, which is not enough to perform analysis with SampEn from Scale 1. Therefore, it is necessary to up-sample by a factor of 10 using the spline interpolation method to get *N_fNIRS_* = 1200 data points, which is enough to analyse with SampEn up to Scale 12. For topographic rendering, the values of MSE EEG and MSE fNIRS at Scale 10 were used for analysis as the mean values with the sufficient size of *N*.

### 2.2. Participants and EEG/fNIRS Recording

The participants of this study were 12 males and 3 females aged from 20 to 30 years old (males: M = 25.08, SD = 2.19; females: M = 25.00, SD = 1.00), all were students of different ethnicities (e.g., Japanese, Chinese, Thai and Indonesian). The participants were self-reported right handed, had normal colour vision with no prior memory disorders and no history of mental illness or sustained physical illness. In line with the study of Alloway, Gathercole and Pickering [47] in which they reported that gender differences do not affect memory performance, this was not considered an exclusionary criterion. 

Concerning the relationship between memory function and age, Lambek and Shevlin [48] suggested that memory growth is constant and begins to level off at the age of twenty. Memory function is also known to decline in later life. Thus, participants within the age range of 20 to 30 years were considered for this study. The participants signed an informed consent form before the experiment and participated in this study voluntarily. The participants were selected based on a purposive sampling method with the criteria shown in Table 1.

The participants were given task-specific instructions as well as a brief practice session to minimise their bodily movements and control their eye movements during the visual task. In line with the study by Yoo and Chung [49], they were seated in a comfortable, height-adjustable chair, at a distance of approximately 50 cm from the computer screen so that their eyes were positioned directly in-line with the fixation point.

As shown in Figure 1, simultaneous EEG/fNIRS data were recorded from cranial locations based on the international 10/20 system using a customised head-cap. In a continuous stimulus set, the participants were asked to look at the screen while each of the 250 indoor/outdoor scenes from the scene understanding (SUN) database [50] were randomly shown. All stimuli were presented on a computer screen for 3 s between eye-fixation on a black-coloured background for 9 s. During this period, the participants were instructed to make a binary choice by clicking the mouse (i.e., the left button to designate “remember” when they believed the test item was part of the memory set and would like to remember it, and the right button to designate “forget”, when they believed the test item was not part of the memory set and would like to forget it). The timing diagram and the sequence of stimuli are shown in Figure 2.

After the measurement, the participants rested for 10 min before performing a scene recognition test. In this part of the task, they had to identify which of the 500 scenes they had previously seen in the experiment. The recognition test scenes comprised 250 scenes from the previous experiment and 250 new ones. 

As noted, Figure 2 illustrates the steps and timing of the experiment. Period *t1* is the prestimulus time before the experiment started. The EEG/fNIRS signals with corrected baseline were then segmented into epochs; each spanning period *t2*, the time for visual stimulus, and *t3*, the time for receiving response from each participant. EEG data were recorded using a Neurofax 1100 (Nihonkoden, Japan) at a sampling frequency of 500 Hz. 

All recorded EEG channels were arranged into a topographic figure. Independent component analysis (ICA) was performed on raw EEG/fNIRS data using the open-source toolbox EEGLAB v12.0.2.5b [51,52] in MATLAB (R2015b; Mathworks Ltd., Natick, MA, USA). Components relating to eye blinking and eye movement artefacts were removed. Other noises were also removed with digital bandpass (0.5 to 50 Hz) and notch (60 Hz) filters. Prestimulus baseline correction was then performed. 

All fNIRS data were recorded by fNIRS Hitachi ETG-7100 (Hitachi, Tokyo, Japan) with a 10 Hz sampling frequency. The discrete wavelet transforms (DWT) was used to remove noise from fNIRS data. The Daubechies 8 wavelet was selected as the mother wavelet. As the epoch interval was 12 s, component d6 was identified as the main task-related component, with side components d5 and d7. Reconstructed signals therefore comprised d5, d6 and d7 components, without the approximated component for baseline removal. This study was approved by the Ethics Committee at Faculty of Information Science and Electrical Engineering, Kyushu University.

For each participant, we examined the brain complexity from the results of the participants’ responses on pictures with visual stimuli. A memory “hit” may be defined as a trial in which the participant chose to remember a visual stimulus in the continuous stimulus set and correctly identified it in the scene recognition test. Conversely, a memory “miss” occurs when the participant chose to remember a picture but could not recognise it in the scene recognition test. Overall numbers of memory hits and misses per participant may be taken to reflect different levels of brain complexity.

After signal reconstruction, all epochs were categorised into four groups: remember and remembered (RR), remember but forgot (RF), forget and forgot (FF) and forget but remembered (FR) cases; the first letter indicating the participants’ preference to remember or forget the stimulus, the second letter indicating their performance (i.e., if they remembered or forgot). The number of epochs in each group therefore depended on the participants’ decisions during the continuous stimulus set and subsequent responses in the scene recognition test. 

From a total of 3750 trials, there were 1163 (31.01%) RR, 348 (9.28%) RF, 1575 (42.00%) FR, 651 (17.36%) FF and 13 (0.35%) no response. All epochs corresponding to the same category at each channel were analysed with the MSE approach.

Note that from the total of 3750 trials, we only considered the complete trials and eliminated the trials that the participants left blank or did not respond to, as shown in Table 2.

## 3. Results

Considering a total of 3750 trials from 15 participants, regardless of whether they wished to remember or forget, the participants remembered 2738 trials or 73.01% of all visual stimuli. This suggests that the brain has the capacity to retain visual information in this experiment no matter how much of an intentional mental state is involved. It indicates the large extent to which memory is under subconscious control.

As shown in Table 3, in terms of pictures viewed per participant, the cohort may be categorised into two groups: Group A (%R1–%R3 at similar levels, such as N0001 and N0003); Group B (%R1–%R3 at different levels, such as N0002 and N0004). 

By splitting the participants in this manner, we hope to potentially gain insights into the application of different strategies or memory-boosting mnemonics. As for %R1 to %R3, they can be calculated from Equations (3)–(5) as follows:%R1 = %RR/(RR + RF)(3)
%R2 = %(RR + FR)/(RR + RF + FR + FF + No response)(4)
%R3 = %FR/(FR + FF)(5)
where, %R1 = the percentage of memory efficiency, %R2 = the percentage of memory recall, %R3 = the percentage of accidental memory recall.

### 3.1. EEG Complexity

In each individual channel, epochs of EEG signals were converted into brain complexity levels using MSE. These were displayed in accordance with levels of sample entropy from Scale 1 to Scale 20. Refer to Equations (1) and (2). 

The average responses from 15 participants are shown in Figure 3. These graphs provide an example of MSE with standard error (SE) bars at electrode site F7. In the frontal lobe, which is associated with WM function, there is a notable gradient between Scale 1 and Scale 3. The remainder of the response exhibits a lower gradient increase in sample entropy.

The study-averaged MSE responses at channel F7 to each of the four epoch classes (RR, RF, FR and FF) are plotted for comparison in Figure 4. This figure illustrates that the RF condition produced the highest level of sample entropy, followed by RR, FR and then FF. 

Furthermore, when combined into two groups: Group 1 (RR and RF), and Group 2 (FR and FF); the findings indicate that attempts to remember (RR and RF) cause greater levels of brain activation than forgetting.

### 3.2. fNIRS Complexity

Similarly to EEG signals, data from 28 fNIRS channels were converted into brain complexity levels using MSE and displayed in terms of sample entropy from Scale 1 to Scale 20. For comparison, graphs of Hb.O2 and Hb.R concentrations converted into brain complexity levels are shown in Figure 5, Figure 6, Figure 7 and Figure 8.

In line with EEG MSE analysis, Figure 5 shows an example of fNIRS MSE with standard error (SE) bars at the F7 site. In comparison with EEG MSE, Hb.O2 from fNIRS MSE is more linear, showing considerably lower levels of entropy due to fNIRS having slow wave characteristics.

Figure 6 shows a graph of fNIRS MSE for Hb.O2 during each epoch type. There is a subtle difference among each of them, with the intention to remember (RR and RF) producing slightly higher levels of fNIRS MSE for Hb.O2.

Figure 7 and Figure 8 show equivalent graphs of fNIRS MSE for Hb.R, displaying similar responses to Hb.O2, both in terms of linearity and sample entropy level. These data mirrored the findings from the EEG MSE results that revealed RF (remember but forgot) produces more frontal brain activity than the RR, FR and FF conditions.

### 3.3. MSE Topography

In order to present a more complete representation of recorded brain activity for each group, topography maps are used to show 30-channel EEG and 28-channel fNIRS data altogether in Figure 9, Figure 10 and Figure 11. These plots are study averages from all participants.

From Scale 1 to Scale 20, the results of MSE taken from Scale 10 or the middle scale are shown in the topography. In this study, Scale 10 is considered acceptable for the sampling frequency in terms of both EEG and fNIRS. 

Figure 9 shows the brain activity presented in levels of entropy from 1.5 to 2.3. The results revealed that the brain activation measured in RR (remember and remembered) is higher when compared to RF, whereas FR and FF are the same. 

Figure 10 and Figure 11 show the topography of fNIRS of Hb.O2 and Hb.R presented in levels of entropy from 0.1 to 0.32, respectively. As shown in Figure 10, for Hb.O2, the results revealed that there is considerable difference between the (RR and RF) and (FR and FF) epoch classes. As for (FR and FF), the brain is more active at the motor cortex and the parietal lobe when compared to (RR and RF).

Figure 11 shows the topography of fNIRS in Hb.R. Despite showing a higher complexity level, it is difficult to differentiate between the four epoch classes. 

### 3.4. Statistical Analysis: Wilcoxon Signed-Rank Test

For statistical analysis, an IBM SPSS Version 22 was used. This study employed the Wilcoxon signed-rank test, a nonparametric statistical technique, to compare a repeated measurement in one sample group to determine the significant difference in mean rank of each pair (i.e., RR-FF, RR-FR, RR-RF, FF-FR, FF-RF and FR-RF). 

The sample size in this study is small (less than 30) so that an assumption of normal distribution is impossible reach. For this case, the use of the Wilcoxon signed-rank test, which is a substitution of paired sample t-test in parametric statistics, is therefore considered as a suitable method of data analysis. 

This study is based on two nominal variables, that is, memory capacity whether the participants remembered or forgot each visual stimulus in the tasks (the experiment and the scene recognition test), and one measurement variable (simultaneous EEG and fNIRS). 

A level of difference in median in memory capacity was significant at zero (W = 0, *p* < 0.05). The results of the Wilcoxon signed-rank test are shown in Table 4.

As mentioned, this section aims to present a significant difference in six pairs of the participants’ responses (i.e., RR-RF, RR-FR, RR-FF, RF-FR, RF-FF and FR-FF) on 30 channels in EEG, each of the 28 channels in Hb.O2 and Hb.R measurements. 

The findings indicate that the responses in the RF group (Md = 0.22) were significantly greater than those of the RR group (Md = 0.17) in channel F7, Z = −2.73, *p* = 0.01, r = 0.50. 

On the other hand, the levels of change in the participants’ brains in the comparison of the responses between the RF and RR groups were equal (Md = 0.20) in channel F3, Z = −2.14, *p* = 0.03, r = 0.39. 

Considering the responses in the pair comparison between the RR and FR groups, the participants’ responses in the RR group (Md = 0.30) showed a significant increase as compared to the responses in the FR group (Md = 0.29) in channel P3, Z = −1.96, *p* = 0.05, r = 0.36. 

Contrary to this, the responses in the RR group (Md = 0.31) manifested a significant decrease when compared with the responses of the FF group (Md = 0.32) in channel FC4, Z = −2.04, *p* = 0.04, r = 0.37. 

As for the pair comparison of the participants’ responses between the FR and RF groups, the findings show that the responses in the RF group demonstrate a significant increase in three different channels as compared to the responses in the FR group, that is,
(i)Channel P3; RF (Md = 0.32) and FR (Md = 0.29), Z = −2.34, *p* = 0.02, r = 0.43;(ii)Channel F5; RF (Md = 0.22) and FR (Md = 0.20), Z = −2.11, *p* = 0.04, r = 0.39;(iii)Channel F7; RF (Md = 0.18) and FR (Md = 0.17), Z = −2.51, *p* = 0.01, r = 0.46.

Besides that, the responses in the RF group (Md = 0.22) were significantly greater than those of the FF group (Md = 0.20) in channel F5, Z = −2.54, *p* = 0.01, r = 0.46. 

To better explain the results from each pair including the significant values, topographical figures are used, see Figure 12, Figure 13 and Figure 14.

## 4. Discussion and Conclusions

While EEG measures the electrical activity of the brain, providing high temporal resolution but low spatial resolution, fNIRS measures the haemodynamic activity of the brain, with comparatively greater spatial resolution and poorer temporal resolution. These two signals complement each other’s weaknesses, such that they may be combined in a common metric (e.g., MSE) to examine underlying cognitive processes.

This study has presented an application of simultaneous EEG and fNIRS recordings, utilising the metric of MSE to index brain complexity levels. This approach may have potential for the investigation of other neurophysiological phenomena underlying brain complexity. Electroencephalography and fNIRS signal coregistration, combining high temporal and spatial resolution, provide a valuable opportunity for studying localised neurocognitive processes. The combination of these two techniques imparts the advantages of each, which compensate for the limitations of the other.

In our population analysis of 3750 individual trials, regardless of whether they wished to remember or forget, the participants remembered a total of 2738 trials or 73.01% of all visual stimuli. This suggests that the brain has the capacity to remember no matter how much of an intentional mental state is involved, indicating the large extent to which memory is under subconscious control.

In this study, both signals were converted to the same entropy scale to allow a comparison of brain activity in a common frame of reference. The maximum level of MSE EEG is approximately 2.3, however the maximum level of MSE fNIRS is nearly 0.3. The level of MSE reveals the characteristics of these two signals (e.g., nonlinearity) which are considerably different. This analysis is considered to be more appropriate than addressing these signals separately in their native format. 

Using neuroimaging to track the patterns of brain activity as well as MSE to analyse the EEG, Hb.O2 and Hb.R data, the results revealed that the RF group showed the highest level followed by the RR, FR and FF groups, respectively. The brain is more active at (RR and RF) levels when compared to (FR and FF). In other words, the intention to remember things is likely to take more mental effort, and therefore more brain activation is shown than when trying to forget them. 

Our research findings are in agreement with several studies, including that of Ang et al. [53] in which the brain is found to be more active during difficult tasks. In this study, RF may be associated with visual image difficulty. This is because some pictures are more difficult to classify, retrieve and recall than others. Consequently, oxy-haemoglobin concentration during this task rises.

In this study, topography figures are used to display synchronised data from every channel. As shown in Figure 12, Figure 13 and Figure 14, there are significant differences found in a number of pairs. However, regarding brain regions related to memory function (i.e., hits), there are significant differences between (RR and RF) at AF4 (EEG), F3 and F7 (Hb.R). As for (RR and FR), there are significant differences at P3 (Hb.R); (RR and FF) at FC4 (Hb.R); (RF and FR) at P3, F5 and F7 (Hb.R); (RF and FF) at AF3 (EEG), FC3 (Hb.O2) and F5 (Hb.R); as well as (FR and FF) at F3 (Hb.O2), *p* < 0.05 (two-tailed). 

As shown in Table 4 and Figure 12, Figure 13 and Figure 14, the brain activation mostly occurred in the left ventrolateral prefrontal cortex (VLPFC), which has been associated with a diverse set of cognitive processes, including actively maintaining information in memory function (AF3, F3, F5, F7 and FC3). 

Our findings appear to agree with Schroeter et al. [19], who performed fNIRS during the Stroop task. They found that differential haemodynamic responses in the left VLPFC correlated with age. It is worth mentioning that these investigations only reported the fNIRS-measured prefrontal cortex activity, whereas the current study measures fast EEG (AF3 and AF4) and slower fNIRS (F3, F5, F7, FC3 and FC4) dominant sites simultaneously using a customised cap. These locations of the frontal lobes are closely related to memory functions, including short-term, working and episodic long-term memory [54].

Aside from the prefrontal cortex, other areas related to memory functions were also monitored. For example, the primary somatosensory cortex (C4 and CP3), which is related to the sense of touch and finger proprioception caused by mouse clicking to respond whether the participants wished to remember or not in the experiment. The angular gyrus in the parietal lobe (P3) is associated with spatial focusing of attention [54], perhaps indicating that the experiment required the participants’ attention. 

Moreover, the primary auditory cortex (T7) and middle temporal gyrus (T8) located in the temporal cortices may have been activated by the operating noise of the fNIRS instrument during the experiment. Furthermore, the secondary visual cortex in the occipital lobe (O2) was shown to be active while participants observed scenery on the display screen [54].

With responses time-locked to the stimuli, the participants demonstrated significantly higher activation in RF when compared to others in the frontal lobe, a region clearly associated with working memory systems and cognitive processing. In agreement with several neuroimaging studies [55,56,57], the brain activation increased in the frontal area. In addition, a combination of all considerations suggests that simultaneous EEG and fNIRS should be preferred to only EEG or fNIRS. In particular, MSE EEG, MSE Hb.O2 and MSE Hb.R can be used for brain analysis.

As a result of the complex and dynamic nature of conscious memory allocation, the participants may have employed unique strategies to memorise selected images. This could have resulted in memory processes overlapping across trials, whereby the participants continuously try to remember a previous stimulus. In this case, the presence of new trials could cause a distraction, which is known to diminish memory performance [54]. The observations support the hypothesis that MSE EEG, MSE Hb.O2 and MSE Hb.R can be used to analyse neuronal activity. This remains a promising finding, and suggests that simultaneous multimodal brain signal recordings of EEG and fNIRS may be advantageous for brain analysis. 

## Figures and Tables

**Figure 1 entropy-22-00189-f001:**
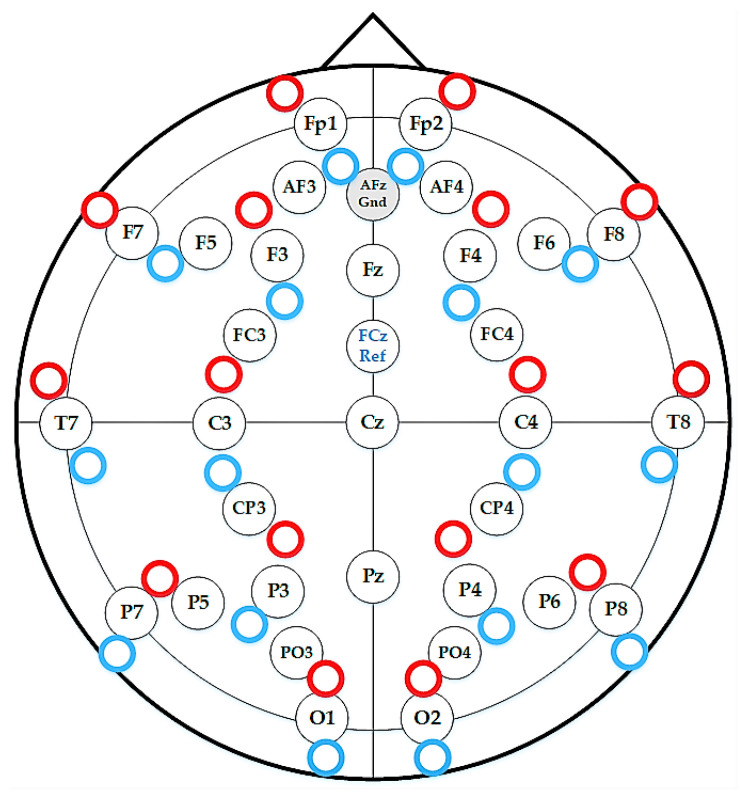
Customised head-cap for simultaneous EEG/fNIRS measurement.

**Figure 2 entropy-22-00189-f002:**
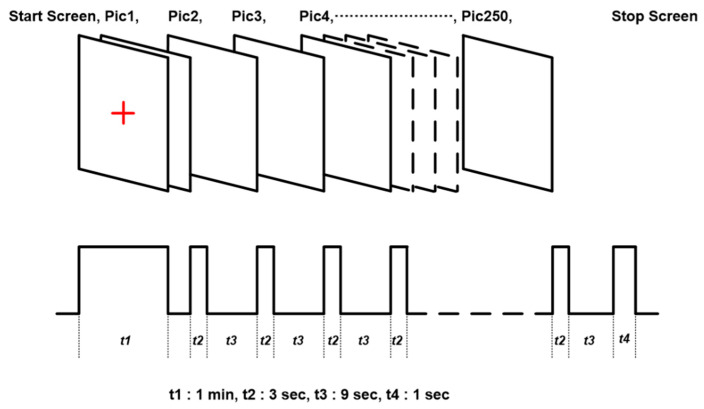
Timing diagram of the cognitive experiment.

**Figure 3 entropy-22-00189-f003:**
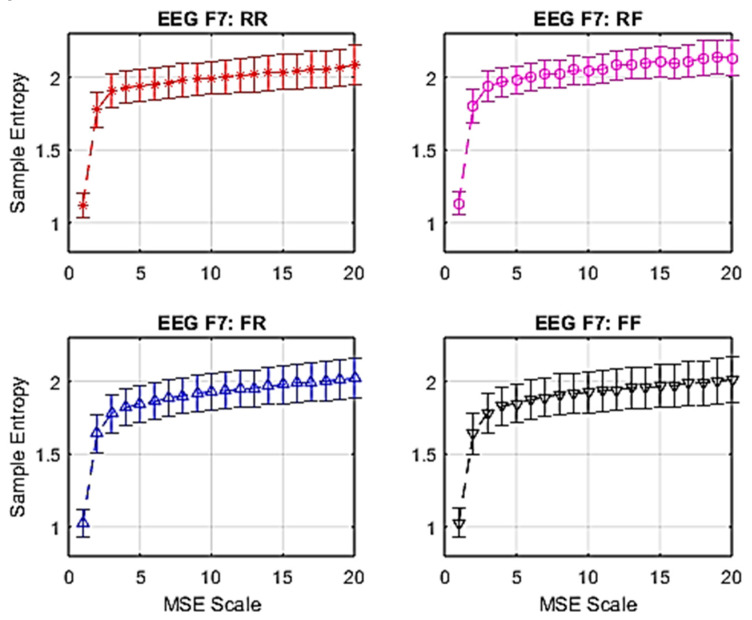
Average multiscale entropy (MSE) of electroencephalogram (EEG) with standard error at F7.

**Figure 4 entropy-22-00189-f004:**
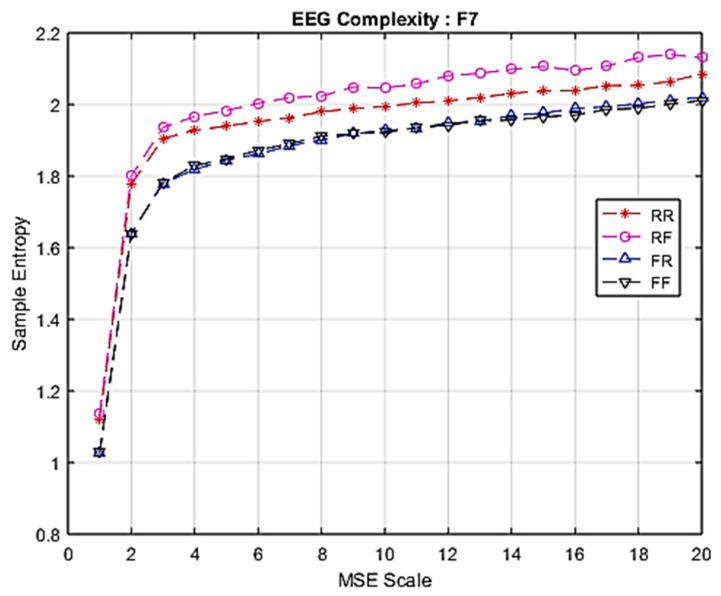
Comparison of EEG complexity among four groups at F7.

**Figure 5 entropy-22-00189-f005:**
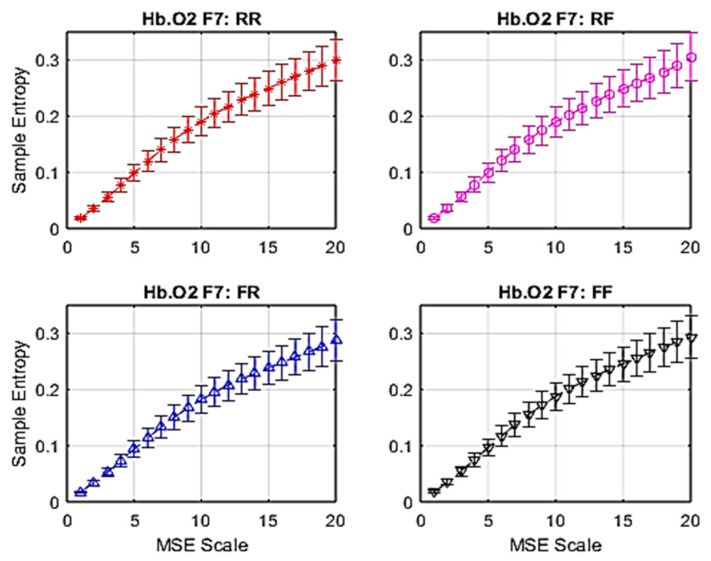
Average MSE of oxy-haemoglobin (Hb.O2) with standard error at F7.

**Figure 6 entropy-22-00189-f006:**
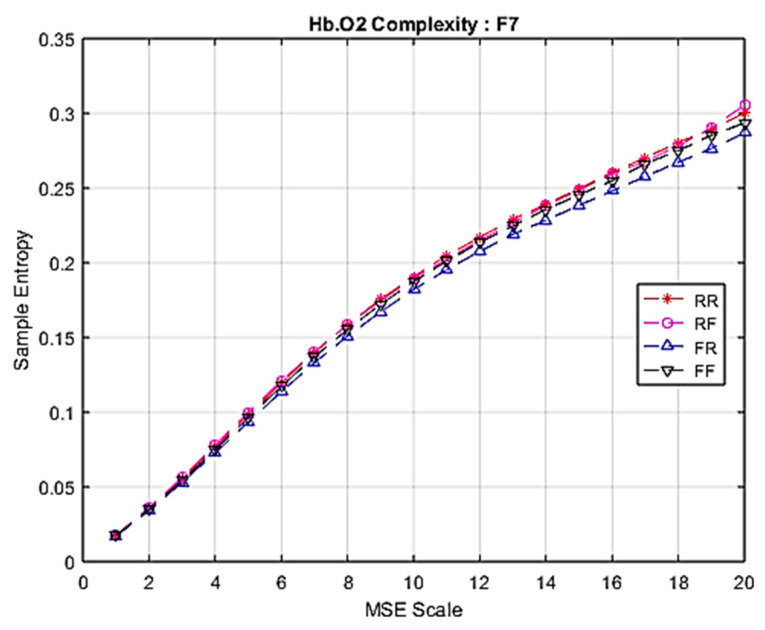
Comparison of Hb.O2 complexity among four groups at F7.

**Figure 7 entropy-22-00189-f007:**
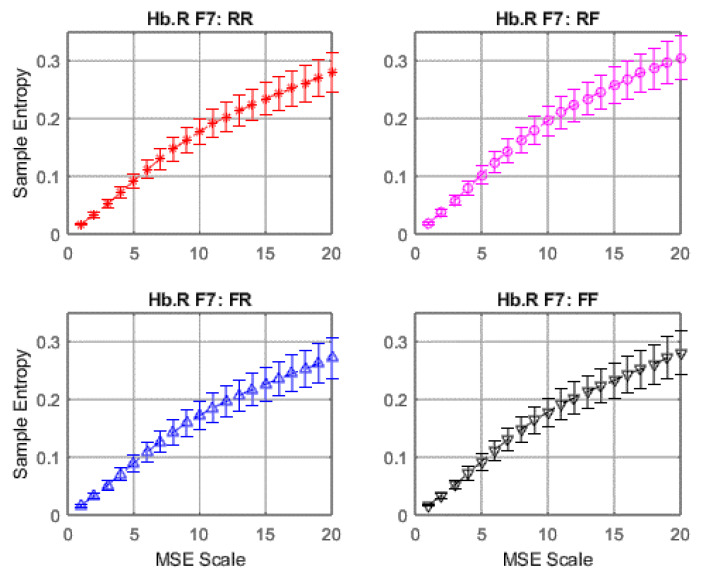
Average MSE of deoxy-haemoglobin (Hb.R) with standard error at F7.

**Figure 8 entropy-22-00189-f008:**
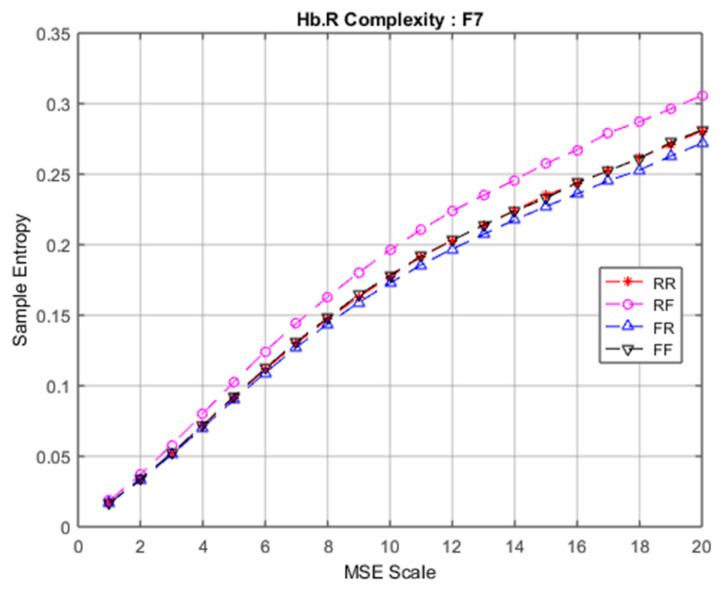
Comparison of Hb.R complexity among four groups at F7.

**Figure 9 entropy-22-00189-f009:**
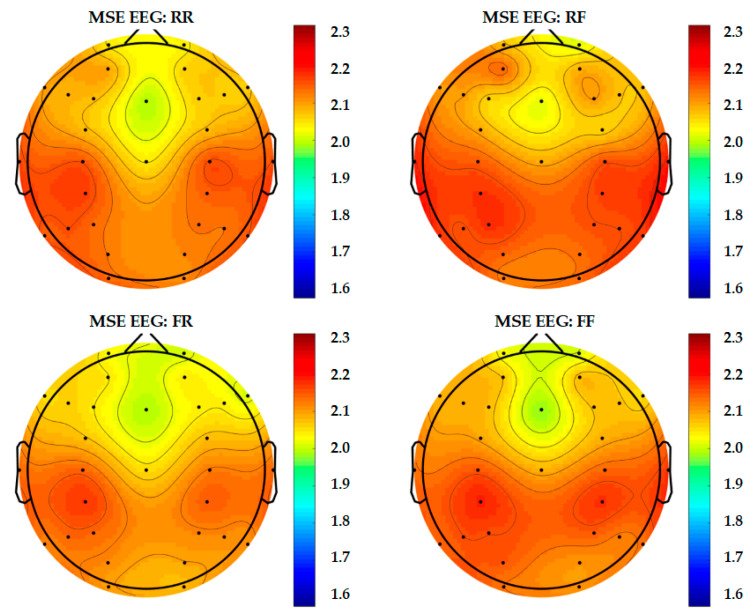
Topography of 30-channel EEG complexity.

**Figure 10 entropy-22-00189-f010:**
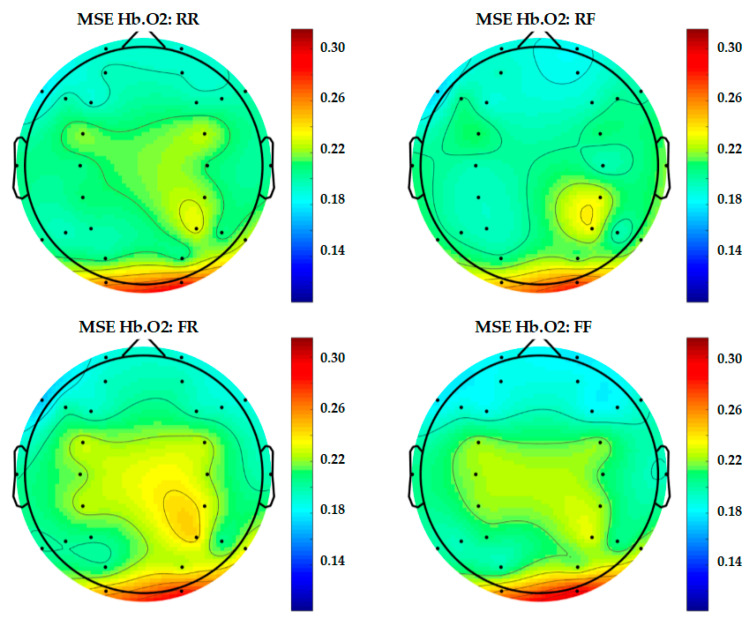
Topography of 28-channel Hb.O2 complexity.

**Figure 11 entropy-22-00189-f011:**
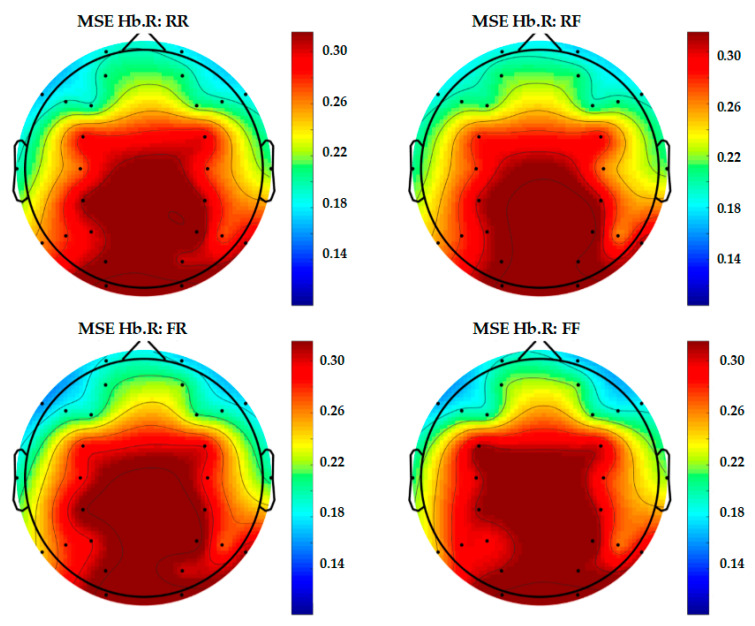
Topography of 28-channel Hb.R complexity.

**Figure 12 entropy-22-00189-f012:**
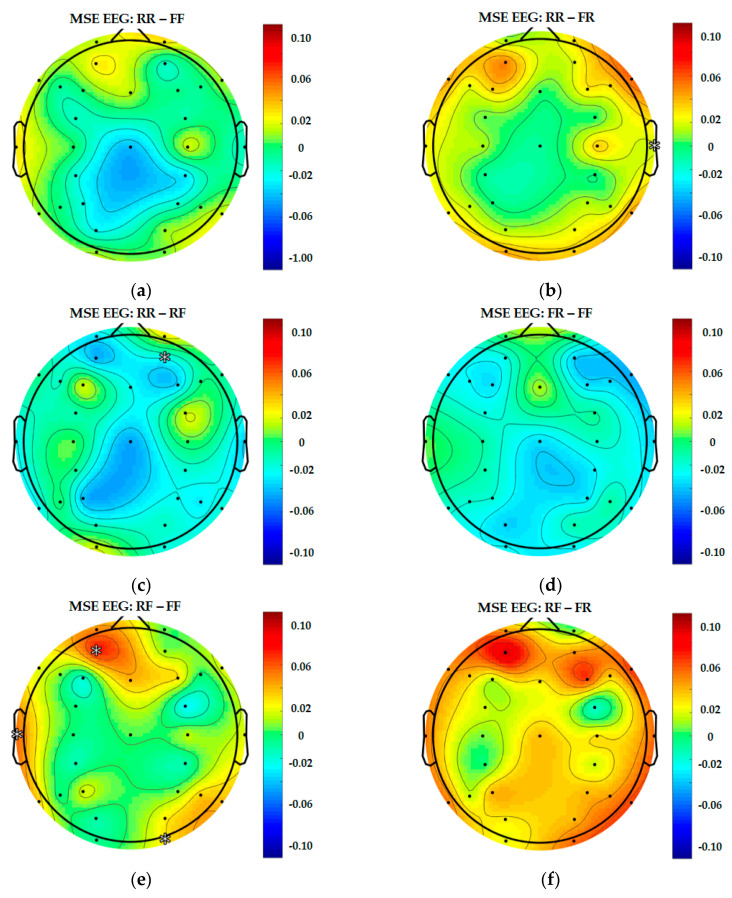
Differentiated results of MSE EEG: (**a**) Differential MSE EEG of RR-FF; (**b**) differential MSE EEG of RR-FR; (**c**) differential MSE EEG of RR-RF; (**d**) differential MSE EEG of FR-FF; (**e**) differential MSE EEG of RF-FF; (**f**) differential MSE EEG of RF-FR.

**Figure 13 entropy-22-00189-f013:**
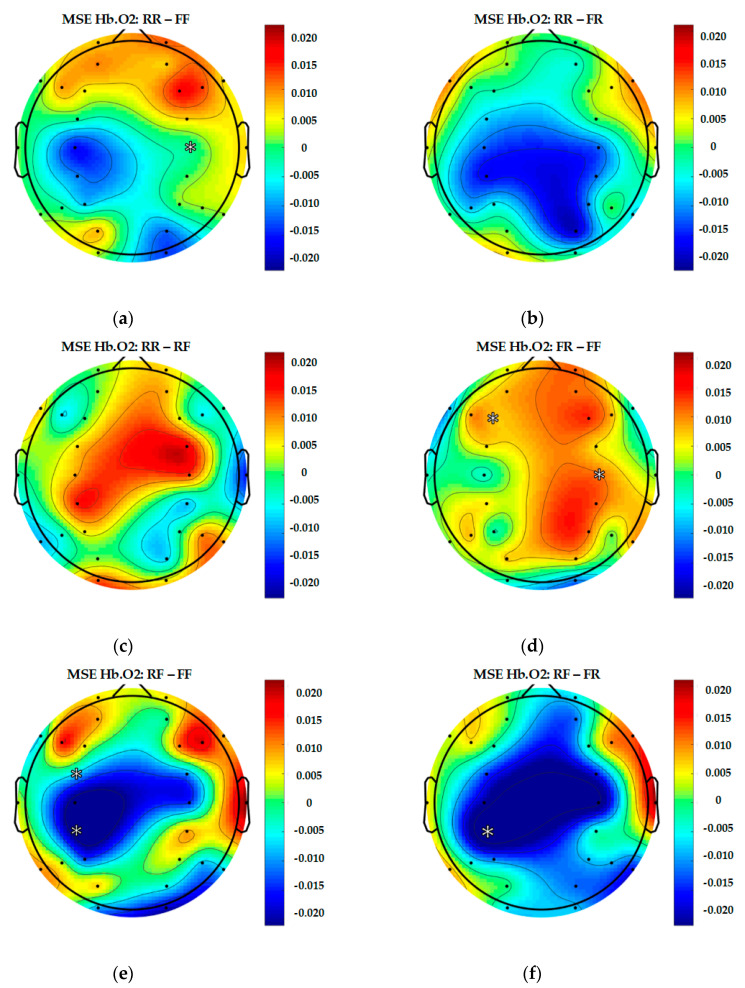
Differentiated results of MSE Hb.O2: (**a**) Differential MSE Hb.O2 of RR-FF; (**b**) differential MSE Hb.O2 of RR-FR; (**c**) differential MSE Hb.O2 of RR-RF; (**d**) differential MSE Hb.O2 of FR-FF; (**e**) differential MSE Hb.O2 of RF-FF; (**f**) differential MSE Hb.O2 of RF-FR.

**Figure 14 entropy-22-00189-f014:**
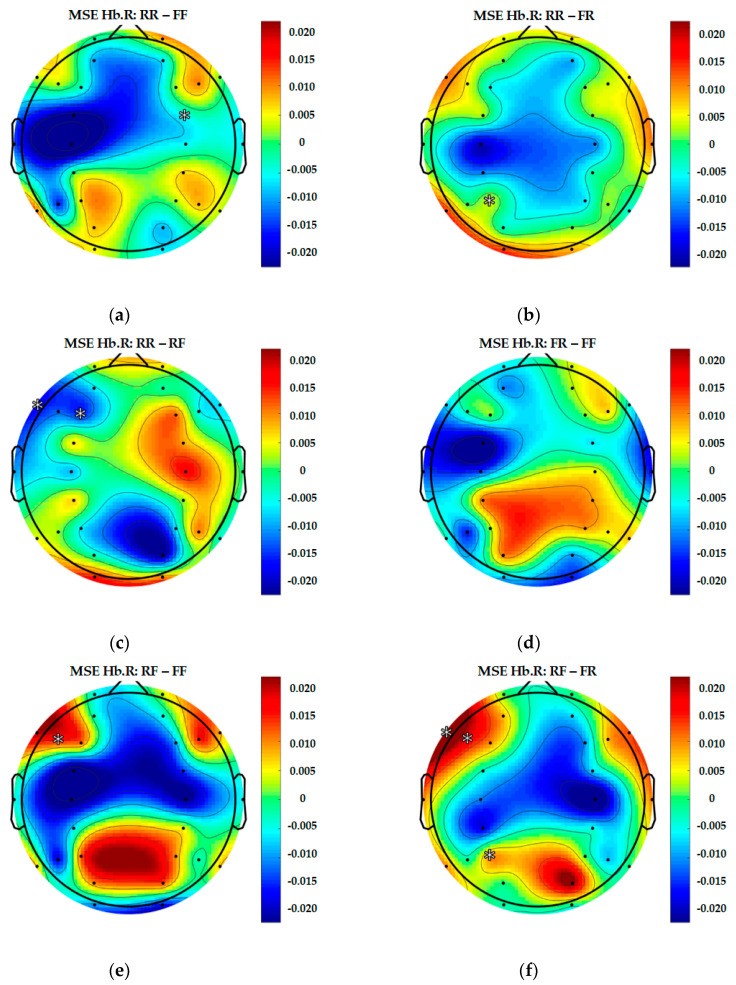
Differentiated results of MSE Hb.R: (**a**) Differential MSE Hb.R of RR-FF; (**b**) differential MSE Hb.R of RR-FR; (**c**) differential MSE Hb.R of RR-RF; (**d**) differential MSE Hb.R of FR-FF; (**e**) differential MSE Hb.R of RF-FF; (**f**) differential MSE Hb.R of RF-FR.

**Table 1 entropy-22-00189-t001:** Selection criteria of participants.

Criteria
1. Age range 20–30
2. Normal colour vision
3. No prior memory disorders
4. No history of mental illness or sustained physical illness

**Table 2 entropy-22-00189-t002:** Demographic information of participants.

Participants ID	Gender	Age	Remember and Remembered (RR)	Remember but Forgot (RF)	Forget but Remembered (FR)	Forget and Forgot (FF)	No Response	Total
N0001	M	21	100	36	86	28	0	250
N0002	F	26	15	1	144	90	0	250
N0003	M	22	68	32	90	59	1	250
N0004	M	26	96	24	58	71	1	250
N0005	F	24	48	12	146	44	0	250
N0006	M	25	74	6	148	22	0	250
N0007	M	25	26	3	193	28	0	250
N0008	M	26	63	25	91	69	2	250
N0009	M	24	103	33	79	34	1	250
N0010	M	28	85	27	94	39	5	250
N0011	M	25	90	32	98	30	0	250
N0012	F	25	112	17	80	41	0	250
N0013	M	29	91	54	68	34	3	250
N0014	M	25	86	17	120	27	0	250
N0015	M	25	106	29	80	35	0	250
Total	M = 12 F = 3	Male (M = 25.08, SD = 2.19) Female (M = 25.00, SD = 1.00)	1163 (31.01%)	348 (9.28%)	1575 (42.00%)	651 (17.36%)	13 (0.35%)	3750 (100%)

**Table 3 entropy-22-00189-t003:** Pictures viewed per participant.

Participant ID	%R1	%R2	%R3	Group
N0001	73.53	74.40	75.44	A
N0002	93.75	63.60	61.54	B
N0003	68.00	63.20	60.40	A
N0004	80.00	61.60	44.96	B
N0005	80.00	77.60	76.84	A
N0006	92.50	88.80	87.06	A
N0007	89.66	87.60	87.33	A
N0008	71.59	61.60	56.88	B
N0009	75.74	72.80	69.91	A
N0010	75.89	71.60	70.68	A
N0011	73.77	75.20	76.56	A
N0012	86.82	76.80	66.12	B
N0013	62.76	63.60	66.67	A
N0014	83.50	82.40	81.63	A
N0015	78.52	74.40	69.57	A
Mean	79.07	73.01	70.11	A = 11
SD	8.92	8.98	11.45	B = 4

**Table 4 entropy-22-00189-t004:** Results of the Wilcoxon signed-rank test.

Types of Measurements	Pairs		Median	Channels	Z	Asymp. Sig. (Two-Tailed)	*r*	Effect Size
EEG	1	RR	2.09	AF4	−2.07 ^b^	0.04	0.38	medium
		RF	2.11					
	2	RR	2.13	T8	−2.23 ^b^	0.03	0.41	medium
		FR	2.15					
	3	RF	2.13	AF3	−1.95 ^b^	0.05	0.36	medium
		FF	2.12					
	4	RF	2.09	O2	−2.21 ^b^	0.03	0.40	medium
		FF	2.09					
	5	RF	2.14	T7	−2.56 ^b^	0.01	0.47	medium
		FF	2.10					
Hb.O2	1	RR	0.23	C4	−1.95 ^b^	0.05	0.36	medium
		FF	0.21					
	2	RF	0.20	CP3	−2.05 ^c^	0.04	0.37	medium
		FR	0.21					
	3	RF	0.25	FC3	−2.29 ^b^	0.02	0.42	medium
		FF	0.23					
	4	RF	0.20	CP3	−2.21 ^c^	0.03	0.40	medium
		FF	0.20					
	5	FR	0.20	F3	−1.98 ^b^	0.05	0.36	medium
		FF	0.18					
	6	FR	0.22	C4	−2.10^b^	0.04	0.38	medium
		FF	0.21					
Hb.R	1	RR	0.20	F3	−2.14^c^	0.03	0.39	medium
		RF	0.20					
	2	RR	0.17	F7	−2.73 ^c^	0.01	0.50	large
		RF	0.22					
	3	RR	0.30	P3	−1.96 ^b^	0.05	0.36	medium
		FR	0.29					
	4	RR	0.31	FC4	−2.04 ^b^	0.04	0.37	medium
		FF	0.32					
	5	RF	0.32	P3	−2.34 ^b^	0.02	0.43	medium
		FR	0.29					
	6	RF	0.22	F5	−2.11 ^b^	0.04	0.39	medium
		FR	0.20					
	7	RF	0.18	F7	−2.51 ^b^	0.01	0.46	medium
		FR	0.17					
	8	RF	0.22	F5	−2.54 ^c^	0.01	0.46	medium
		FF	0.20					

Note. ^b^ Based on negative ranks; ^c^ Based on positive ranks.

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
