# Peer review of "Multiscale Entropy as a New Feature for EEG and fNIRS Analysis"

_entropy, 2020, doi:10.3390/e22020189_

Round 1

Reviewer 1 Report

The following shortcomings must be addressed.

I was excited when I first read the title thinking the manuscript was on a multimodal method, combining EEG and NIRS; however, this is not the case. As the two modalities are considered separately, the authors have not clearly presented how the two are complementing each other in their results and conclusions considering their intrinsic characteristics. Results presented in Table 1 and Figure 12 need to be discussed in detail from an anatomical perspective, including how they corroborate with results from other studies. What values were chosen for the template length (m) and tolerance (r) to obtain the MSE results shown in Figures 3 – 8? What is the justification for the chosen values? Are the results invariant to changes in these values? The complete demographics of the participants must be listed. The authors noted that, “it is a deliberate choice of participants due the qualities they possesses” (line 179).  What are these qualities? Larger font size needs to be used in Figures 3 – 12. Clarity of these figures are not good, especially on printed copies of the manuscript. References cited and listed need to carefully checked. There seem to be a number of errors. For example, Refs. 18 (line 88), 41 (line 190), and 42 (line 358) do not appear to be correct.

The following are a few nitpicking items.

It is not clear how past and present events can be linked with future events, which by definition have not yet occurred (line 34). What were the EEG and fNIRS sampling frequencies? The trials for RR, RF, FR and FF don’t add up to 3,750 (lines 231-232). What trials were excluded and why? Table 1 should include a caption. The second paragraph on pg. 13 (lines 348-352) is misleading. It is better to discuss in terms of pictures viewed per participant rather than the total number by all participants.

Author Response

Dear Reviewer1,

The revision of manuscript has been attached. Thank you.

Response to Reviewer 1 Comments

Point 1: As the two modalities are considered separately, the authors have not clearly presented how the two are complementing each other in their results and conclusions considering their intrinsic characteristics.

 Response 1:

 Thank you very much for your suggestion, the results and conclusions have been improved and stated more clearly how EEG and fNIRS can complement each other as follows.

(Lines 443-513)

While EEG measures electrical activity from the brain, providing high temporal resolution but low spatial resolution, fNIRS measures haemodynamic activity of the brain, with comparatively greater spatial resolution and poorer temporal resolution. These two signals complement each other’s weaknesses, such that they may be combined in a common metric (e.g. MSE) to examine underlying cognitive processes.

This study has presented an application of simultaneous EEG and fNIRS recordings and utilising the metric of MSE to index brain complexity level. This approach may have prospects for the further investigation of other neurophysiological phenomena underlying brain complexity. Electroencephalography and fNIRS signal co-registration, combining high temporal and spatial resolution, provides a valuable opportunity for studying localised neurocognitive processes. The combination of these two techniques imparts the advantages of each, which compensate for the limitations of the other.

For the population analysis of 3,750 trials, regardless of whether they wished to remember or forget, the participants remembered a total of 2,738 trials or 73.01% of all visual stimuli. This suggests that the brain has the capacity to remember no matter how much an intentional mental state is involved, indicating the large extent to which memory is under subconscious control.

In this study, both signals were converted to the same entropy scale to allow a comparison of brain activity in a common frame of reference. The maximum level of MSE EEG is approximately at 2.3, however the maximum level of MSE fNIRS is nearly 0.3. The level of MSE reveals characteristics of these two signals (e.g. non-linearity) that are considerably different. This analysis is considered to be more appropriate that addressing these signals separately in their native format.

Using neuroimaging to track patterns of brain activity as well as MSE to analyse the EEG, Hb.O2, and  Hb.R data, the results revealed that RF shows the highest level followed by RR, FR, and FF, respectively. The brain is more active at (RR and RF) levels when compared to (FR and FF). In other words, the intention to remember things is likely to take more mental effort, and therefore show more brain activation than trying to forget them.

Our research findings agree with several studies, including Ang et al. [52], in which brain is found to be more active during difficult tasks. In this study, (RF) may be associated with visual image difficulty. This is because some pictures are more difficult to classify, retrieve, and recall than others. Consequently, oxy-haemoglobin concentration during this task rises.

In this study, topography figures are used to display synchronized data from every channel. As shown in Figures 12 – 14, there are significant differences found in a number of pairs. However, regarding brain regions related to memory function (i.e. hits), taking this into consideration, there are significant differences between (RR and RF) at AF4 (EEG), F3 and F7 (Hb.R). As for (RR and FR), there are significant differences at P3 (Hb.R), (RR and FF) at FC4 (Hb.R), (RF and FR) at P3, F5 and F7 (Hb.R), (RF and FF) at AF3 (EEG), FC3 (Hb.O2) and F5 (Hb.R) as well as (FR and FF) at F3 (Hb.O2), p < 0.05 (two-tailed).

As shown in Table 4 and Figures 12 – 14, the brain activation mostly occurred in left ventrolateral prefrontal cortex (VLPFC), which has been associated with a diverse set of cognitive processes, including actively maintaining information in memory function (AF3, F3, F5, F7 and FC3).

Our findings appear to agree with Schroeter et al. [19], who performed fNIRS during the Stroop task. They found differential haemodynamic responses in the left VLPFC correlated with age. It is worth mentioning that the investigations reported only fNIRS-measured prefrontal cortex activity, whereas the current study measures fast EEG (AF3 and AF4) and slower fNIRS (F3, F5, F7, FC3 and FC4) dominant sites simultaneously using a customized cap. These locations of the frontal lobes are closely related to memory functions, including short-term, working, and episodic long-term memory [53].

Aside from the prefrontal cortex, other areas related to memory functions were also monitored. For example, the primary somatosensory cortex (C4 and CP3), which is related to the sense of touch and finger proprioception caused by mouse clicking to respond whether the participants wished to remember or not in the experiment.The angular gyrus in the parietal lobe (P3) is associated with spatial focusing of attention [53], perhaps indicating that experiment required the participants’ attention.

Moreover, the primary auditory cortex (T7) and middle temporal gyrus (T8) located in the temporal cortices may have been activated by operating noise of the fNIRS instrument during the experiment. Furthermore, the secondary visual cortex in the occipital lobe (O2) was shown to be active while participants observed scenery on the display screen [53].

With responses time-locked to the stimuli, the participants demonstrated significantly higher activation in (RF) when compared to others in the frontal lobe, a region clearly associated with working memory systems and cognitive processing. In agreement with several neuroimaging studies [54,55,56], the brain activation increased in the frontal area. In addition, a combination of all considerations suggests that simultaneous EEG and fNIRS should be preferred to only EEG or fNIRS. In particular, MSE EEG, MSE Hb.O2 and MSE Hb.R can be used for brain analysis.

Due to the complex and dynamic nature of conscious memory allocation, the participants may have employed unique strategies to memorise selected images. This could have resulted in memory processes overlapping across trials, whereby the participants continuously try to remember a previous stimulus. In this case, the presence of new trials could cause a distraction, which is known to diminish memory performance [53]. The observations support the hypothesis that MSE EEG, MSE Hb.O2 and MSE Hb.R can be used to analyse neuronal activity. This remains a promising finding, and suggests that simultaneous multimodal brain signal recordings of EEG and fNIRS may be advantageous for brain analysis.

Point 2: Results presented in Table 1 and Figure 12 need to be discussed in detail from an anatomical perspective, including how they corroborate with results from other studies.

Response 2:

Thank you very much for your constructive suggestion.

Note: Table 1 has been changed to Table 4 because of additional information added to Tables 2-3. Figure 12 has been adjusted to be Figure 12, Figure 13 and Figure 14 to improve the clarity of figures.

 (Lines 480-499)

As shown in Table 4 and Figures 12 – 14, the brain activation mostly occurred in left ventrolateral prefrontal cortex (VLPFC), which has been associated with a diverse set of cognitive processes, including actively maintaining information in memory function (AF3, F3, F5, F7 and FC3).

Our findings appear to agree with Schroeter et al. [19], who performed fNIRS during the Stroop task. They found differential haemodynamic responses in the left VLPFC correlated with age. It is worth mentioning that the investigations reported only fNIRS-measured prefrontal cortex activity, whereas the current study measures fast EEG (AF3 and AF4) and slower fNIRS (F3, F5, F7, FC3 and FC4) dominant sites simultaneously using a customized cap. These locations of the frontal lobes are closely related to memory functions, including short-term, working, and episodic long-term memory [53].

Aside from the prefrontal cortex, other areas related to memory functions were also monitored. For example, the primary somatosensory cortex (C4 and CP3), which is related to the sense of touch and finger proprioception caused by mouse clicking to respond whether the participants wished to remember or not in the experiment.The angular gyrus in the parietal lobe (P3) is associated with spatial focusing of attention [53], perhaps indicating that experiment required the participants’ attention.

Moreover, the primary auditory cortex (T7) and middle temporal gyrus (T8) located in the temporal cortices may have been activated by operating noise of the fNIRS instrument during the experiment. Furthermore, the secondary visual cortex in the occipital lobe (O2) was shown to be active while participants observed scenery on the display screen [53].

Point 3: What values were chosen for the template length (m) and tolerance (r) to obtain the MSE results shown in Figures 3 – 8? What is the justification for the chosen values? Are the results invariant to changes in these values? 

Response 3:

Thank you very much. Based on your useful suggestion, the manuscript has been improved as follows.

(Lines 182-195)

Previous studies have shown that SampEn has the acceptable range for m= 2, r= 0.15 x SD of the time series [38,46]. Adopting the method proposed by Catarino et al. [46], we used m = 2, r = 0.15 x SD of the time series. For N, we calculated from the sampling frequency and epoch length used for quantitative analysis (12 seconds). According to Richman and Moorman [38], the appropriate N size for SampEn analysis must be a minimum of 10m to 20m, where m = 2; N should be at least 100 to 400 in each scale. When the scale increases, the number of N in each scale will decrease. For EEG sampling frequency at 500 Hz, when calculating, the size of N, NEEG is equal to 500 Hz x 12 sec = 6000 data points. Considering the 20th scale, the size of the NEEG is 300, which is still enough for SampEn analysis. With respect to fNIRS, which has a sampling frequency of only 10 Hz. To calculate the size of N, NfNIRS = 10 Hz x 12 sec = 60 data points, which is not enough to perform analysis with SampEn from Scale 1. Therefore, it is necessary to up-sample by a factor of 10 using the spline interpolation method to get NfNIRS = 1200 data points, which is enough to analyse with SampEn up to Scale 12. For topographic rendering, the values of MSE EEG and MSE fNIRS at Scale 10 were used for analysis as the mean values with the sufficient size of N.

Point 4: The complete demographics of the participants must be listed. The authors noted that, “it is a deliberate choice of participants due the qualities they possesses” (line 179). What are these qualities? 

Response 4:

Thank you very much for your expertise. The manuscript has been improved in the methodology section. The demographic of the participants in has been listed, the selection criteria of participants has been added, and Line 179 has been adjusted and improved as follows.

(Lines 207-227)

The participants of this study were 12 males and 3 females aged from 20 to 30 years old (males: (M=25.08, SD=2.19); females :(M=25.00, SD=1.00), all were students of different ethnicities (e.g. Japanese, Chinese, Thai and Indonesian). The participants had self-reported right handed, normal colour vision with no prior memory disorders and no history of mental illness or sustained physical illness. In line with the study of Alloway, Gathercole, and Pickering [47], in which they reported that gender differences do not affect memory performance, this was not considered an exclusionary criterion. Concerning the relationship between memory function and age, Lambek and Shevlin [48], suggested that memory growth is constant and begins to level off at the age of twenty. Memory function is also known to decline in later life. Thus, the participants within the age range of 20 to 30 years were considered for this study. The participants signed on the informed consent before the experiment and participated in this study voluntarily. The participants were selected based on a purposive sampling method with the following criteria shown in Table 1.

Table 1. Selection criteria of participants

Criteria

1. Range of age between 20-30

2. Normal colour vision

3. No prior memory disorders

4. No history of mental illness or sustained

  physical illness

The participants were given task-specific instructions as well as a brief practice session to minimise their bodily movements and control their eye movements during the visual task. In line with a study by Yoo and Chung [49], they were seated in a comfortable height-adjustable chair, at a distance of approximately 50 cm from the computer screen so that their eyes were positioned directly in-line with the fixation point

(Lines 273-292)

 4.1 Demographic Information of Participants

After signal reconstruction, all epochs were categorised into four groups: remember and remembered (RR), remember but forgot (RF), forget and forgot (FF), and forget but remembered (FR) cases; the first letter indicating the participants' preference to remember or forget the stimulus, the second letter indicating their performance (i.e. if they remembered or forgot). The number of epochs in each group therefore depended on the participants' decisions during the continuous stimulus set and subsequent responses in the scene recognition test. From a total of 3,750 trials, there were 1,163 (31.01%) RR, 348 (9.28%) RF, 1,575 (42.00%) FR, 651 (17.36%) FF, and 13 (0.35%) no response. All epochs corresponding to the same category at each channel were analysed with the MSE approach.

Note that, from the total of 3,750 trials, we considered only the complete trials and eliminated the trials that the participants left blank or did not respond from the study as shown in Table 2.

Table 2. Demographic information of participants

Participants ID

Gender

Age

RR

RF

FR

FF

No Response

Total

N0001

M

21

100

36

86

28

0

250

N0002

F

26

15

1

144

90

0

250

N0003

M

22

68

32

90

59

1

250

N0004

M

26

96

24

58

71

1

250

N0005

F

24

48

12

146

44

0

250

N0006

M

25

74

6

148

22

0

250

N0007

M

25

26

3

193

28

0

250

N0008

M

26

63

25

91

69

2

250

N0009

M

24

103

33

79

34

1

250

N0010

M

28

85

27

94

39

5

250

N0011

M

25

90

32

98

30

0

250

N0012

F

25

112

17

80

41

0

250

N0013

M

29

91

54

68

34

3

250

N0014

M

25

86

17

120

27

0

250

N0015

M

25

106

29

80

35

0

250

Total

M = 12

F = 3

Male (M=25.08, SD=2.19)

Female

(M=25.00, SD=1.00)

1163

(31.01%)

348

(9.28%)

1575

(42.00%)

651

(17.36%)

13

(0.35%)

3750

(100%)

Considering a total of 3,750 trials from 15 participants, regardless of whether they wished to remember or forget, the participants remembered 2,738 trials or 73.01% of all visual stimuli. This suggests that the brain has the capacity to retain visual information in this experiment no matter how much an intentional mental state is involved. This indicates the large extent to which memory is under subconscious control.

Point 5: Larger font size needs to be used in Figures 3 – 12.

Response 5:

The font size in Figures 3-12 has been adjusted, thank you.

Point 6: Clarity of these figures are not good, especially on printed copies of the manuscript.

Response 6:

The clarity of the figures has been improved. We hope these are now acceptable. Thank you.

Point 7: References cited and listed need to carefully checked. There seem to be a number of errors. For example, Refs. 18 (line 88), 41 (line 190), and 42 (line 358) do not appear to be correct.

Response 7:

References cited and listed have been corrected. Thank you for highlighting these.

Point 8: It is not clear how past and present events can be linked with future events, which by definition have not yet occurred (line 34).

Response 8:

Thank you very much for highlighting these. The manuscript has been improved as follows.

(Lines 114-117)

Memory plays a crucial role in human life. It allows us to construct life-narratives by linking past, present and future events. Past events that we have experienced are important because they enable us to make educated decisions in the present and to construct mental scenarios in real-time; this forward-thinking, or ability to simulate future events, is a quintessentially human trait [25].

Point 9: What were the EEG and fNIRS sampling frequencies?

Response 9:

Thank you very much for highlighting these. The manuscript has been improved as follows.  

(Lines 187-190)

For EEG sampling frequency at 500 Hz, when calculating, the size of N, NEEG is equal to 500 Hz x 12 sec = 6000 data points. Considering the 20th scale, the size of the NEEG is 300, which is still enough for SampEn analysis. With respect to fNIRS, which has a sampling frequency of only 10 Hz.

Point 10: The trials for RR, RF, FR and FF don’t add up to 3,750 (lines 231-232). What trials were excluded and why? Table 1 should include a caption.

Response 10:

 Thank you very much for highlighting these. The manuscript has been improved as follows.

(Lines 282-292)

Note that, from the total of 3,750 trials, we considered only the complete trials and eliminated the trials that the participants left blank or did not respond from the study as shown in Table 2.

Table 2. Demographic information of participants

Participants ID

Gender

Age

RR

RF

FR

FF

No Response

Total

N0001

M

21

100

36

86

28

0

250

N0002

F

26

15

1

144

90

0

250

N0003

M

22

68

32

90

59

1

250

N0004

M

26

96

24

58

71

1

250

N0005

F

24

48

12

146

44

0

250

N0006

M

25

74

6

148

22

0

250

N0007

M

25

26

3

193

28

0

250

N0008

M

26

63

25

91

69

2

250

N0009

M

24

103

33

79

34

1

250

N0010

M

28

85

27

94

39

5

250

N0011

M

25

90

32

98

30

0

250

N0012

F

25

112

17

80

41

0

250

N0013

M

29

91

54

68

34

3

250

N0014

M

25

86

17

120

27

0

250

N0015

M

25

106

29

80

35

0

250

Total

M = 12

F = 3

Male (M=25.08, SD=2.19)

Female

(M=25.00, SD=1.00)

1163

(31.01%)

348

(9.28%)

1575

(42.00%)

651

(17.36%)

13

(0.35%)

3750

(100%)

Considering a total of 3,750 trials from 15 participants, regardless of whether they wished to remember or forget, the participants remembered 2,738 trials or 73.01% of all visual stimuli. This suggests that the brain has the capacity to retain visual information in this experiment no matter how much an intentional mental state is involved. This indicates the large extent to which memory is under subconscious control.

Point 11: The second paragraph on pg. 13 (lines 348-352) is misleading. It is better to discuss in terms of pictures viewed per participant rather than the total number by all participants.

Response 11:

Thank you very much for highlighting these. The manuscript has been improved as follows.

(Lines 294-312)    

We included Table 3 for discuss in terms of pictures viewed per participant as follows.

Table 3. Pictures viewed per participant

Participant

ID

%R1

%R2

%R3

Group

N0001

73.53

74.40

75.44

A

N0002

93.75

63.60

61.54

B

N0003

68.00

63.20

60.40

A

N0004

80.00

61.60

44.96

B

N0005

80.00

77.60

76.84

A

N0006

92.50

88.80

87.06

A

N0007

89.66

87.60

87.33

A

N0008

71.59

61.60

56.88

B

N0009

75.74

72.80

69.91

A

N0010

75.89

71.60

70.68

A

N0011

73.77

75.20

76.56

A

N0012

86.82

76.80

66.12

B

N0013

62.76

63.60

66.67

A

N0014

83.50

82.40

81.63

A

N0015

78.52

74.40

69.57

A

Mean

79.07

73.01

70.11

A = 11

SD

8.92

8.98

11.45

B = 4

Where  %R1 = the percentage of memory efficiency

           %R2 = the percentage of memory recall

           %R3 = the percentage of accidental memory recall

           %R1 to %R3 were calculated from equations 4 – 6 as follows.

%R1=%RR/(RR+RF)                                                      (4)

%R2 = %(RR+FR)/(RR+RF+FR+FF+No response)            (5)

%R3 = %FR/(FR+FF)                                                     (6)

As shown in Table 3, in terms of pictures viewed per participant, the cohort may be categorised into two groups: Group A (%R1 - %R3 at similar levels such as N0001 and N0003); Group B (%R1 - %R3 at different levels such as N0002 and N0004). By splitting the participants in this manner, we hope to potentially gain insights into the application of different strategies or memory-boosting mnemonics.

...........................................End of Response to Reviewer...........................

Dear Reviewer1,

First of all, we would like to thank you for your time and kind suggestion. We really appreciate and feel honoured. This study is not all plain sailing as no part came smoothly, effortlessly and painlessly. Some of the challenges were managerial issues, while others were research-based in nature. Nonetheless, there were some significant and exciting findings emerged from this study. We believe it is worth sharing so that other researchers could benefit from this study. We truly hope this manuscript will be helpful for other researchers if it is made publicly available.

Secondly, we have improved the manuscript based on your suggestion and comments. However, due to the major revision, this manuscript has totally been adjusted and improved. For this reason, we could not provide track changes or highlights as expected because this revision has been adjusted to a totally new, improved one. We have tried to use track changes for response to reviewers, but it is very difficult to read, spot and understand.  Therefore, in response to reviewers, we have provided all (including lines and sentences) in the form of “Point and Response to reviewers” separately into each point to improve and hopefully it will easily be read and spotted. Please find the attachment of “Responses to Reviewers”.

Finally, it would be really appreciated if you could provide the verification before 4th January. My name is Thanate Angsuwatanakul, the first author of this manuscript. I am currently a PhD student, Kyushu University, and now in the process of VIVA examination. The date of VIVA examination is on 6th January 2020 at 3.30 p.m. However, I am still not qualified to attend the exam until I submit the proof that my manuscript is accepted for publication in reputable refereed journals. The main problem is my candidature term end is 10th January 2020 meaning that I have about one week for the process of VIVA examination. I realize that this situation is very difficult not only for me but for everyone including the reviewers for kind assistance to verify this manuscript. Honestly, I realize that this manuscript may be rejected from the reviewers meaning that I will be terminated very soon. However, I will not regret because my intention is I would like to submit (publish) in “Entropy” once in my life and I have already done my best. 

Thank you very much for your kind assistance and consideration. I look forward to hearing from you soon.

Sincerely yours,

Thanate Angsuwatanakul

Reviewer 2 Report

The paper proposes to use multiscale entropy (MSE) as a new feature for brain analysis using EEG and fNIRS. Comments and suggestions are given below.

Section 1: Introduction.

     This section starts with a literature review of memory, brain and current studies on understanding of memory functions based on neuroimaging modalities such as fNIRS, fMRI, EEG and MEG. The review then focuses on the difference between EEG and fNIRS signals in measuring the brain complexity. Simultaneous EEG - fNIRS for measuring brain complexity is also reviewed. The paper then jumps to proposing multiscale entropy (MSE) to analyse EEG - fNIRS data.

     The literature review is lengthy and cannot point out the key problem, gap, disadvantage or limitation of the current studies that needs to be addressed. The paper does not explain why entropy is needed, why MSE is proposed to analyse the EEG - fNIRS data, and why other neuroimaging modalities are not chosen.

Section 2: MultiScale Sample Entropy (MSE).

     The abbreviation MSE has been used for multiscale entropy, it is now used for multiscale sample entropy although multiscale sample entropy and multiscale entropy are not the same.

     Different entropy functions in the literature are reviewed in a very short paragraph and then Sample Entropy is introduced for multiscale entropy. It is not clear why those entropy functions cannot be applied, and why only Sample Entropy is chosen.

     A theoretical background on entropy and a comparison between entropy functions are required.

     The sentence on lines 154 and 155 “where m is template length, r is the tolerance for accepting matches, N is the number of data points, Ai is the number of matches of length m+1, and Bi is number of matches of length m” needs to be revised (remove indentation, correct subscript for Ai and Bi, and use italic font for m, N, Ai, and Bi). The same for the sentence on lines 160 and 161.

Section 3:  Materials and Methods.

     The research design is good. The number of participants is small.

     It would be helpful for other researchers to use the collected data set if it is made publicly available.

Section 4: Results.

     The multiscale entropy was already used as seen in references 35-38. However, there are no results for comparison of the proposed multiscale entropy in this paper with the current one in those references.

Section 5: Discussion and Conclusions.

      The research findings and discussions are interesting.

Author Response

Dear Reviewer2,

The revision of manuscript has been attached. Thank you.

Response to Reviewer 2 Comments

Section I: Introduction

Point 1: This section starts with a literature review of memory, brain and current studies on understanding of memory functions based on neuroimaging modalities such as fNIRS, fMRI, EEG and MEG. The review then focuses on the difference between EEG and fNIRS signals in measuring the brain complexity. Simultaneous EEG - fNIRS for measuring brain complexity is also reviewed. The paper then jumps to proposing multiscale entropy (MSE) to analyse EEG - fNIRS data.

Response 1:

Thank you very much for your suggestion.   We admitted that the literature review was not well-organized and might be misleaded from the title of this manuscript. In the revision, we reorganize the story in mainbody and abstract to be EEG/fNIRS oriented perspective, especially in the study of brain complexity using this two modalities, and address memory task as a potential application of this multimodality approach.

(Lines 35 – 53)

Each neuroimaging modality has its own peculiarities, pros and cons. For example, EEG gives high temporal resolution but low spatial resolution, whereas fNIRS offers high spatial resolution but relatively low temporal resolution (compared with EEG). Furthermore, fNIRS does not suffer from electromyographic (EMG) and blink artifacts. Combining the two modalities of EEG and fNIRS is therefore a complementary approach for brain function analysis with the strengths of each method supporting the weaknesses of the other. The simultaneous acquisition of EEG and fNIRS may therefore provide a more complete picture of brain activity than either of the two individual modalities. In this sense, we believe that fNIRS should not be considered as a competitor of EEG, but rather as a pooling modality for cases in which one technique is insufficient.

For an in-depth understanding and insight of what is happening inside the human brain, both high spatial and high temporal resolution are likely to be beneficial. Research studies combining simultaneous EEG-fNIRS to examine memory functions are scant. However, there are a handful of published studies using simultaneous EEG and fNIRS to assess visual memory function [1,2,3] employed bimodal recordings of EEG and fNIRS to measure and discriminate among levels of mental workload; reporting promising results for the accurate quantification of mental workload in real-world settings. In addition, Sun et al. [4] proposed a multimodal approach to evaluate EEG and fNIRS signals for affective state detection. Their results indicated that simultaneous EEG and fNIRS improves performance over fNIRS or EEG only. Despite using bimodal recordings of EEG and fNIRS, neither of these studies have explored the application of MSE for signal analysis.

(Lines 54-149)

1.1. EEG as Measurement for Brain Complexity

Electroencephalography  is  regarded  as  a  non-invasive,  functional neuroimaging technique [5], which consists of measuring electrical activity from the brain using electrodes

placed on the scalp. This has been applied widely in order to investigate neural activity underlying these processes. Since EEG can measure brain activity with high temporal resolution, it is suitable for capturing dynamic brain activity. Recording EEG remains the most effective technique for observing near-instantaneous neural responses to visual stimuli [6].

Furthermore, EEG devices are inexpensive and robust for the measurement of brain activity [7]. However, the fundamental limitation of EEG is low spatial resolution due to the diffusion of electrical signals travelling between their intracranial source and electrodes on the scalp. The inverse square law of electric fields also limits the influence of sub-cortical structures on EEG waveforms. Generally, theta (4–7 Hz) and alpha (8–12 Hz) frequency ranges are dominant during working memory tasks [8]. Increases in theta range spectral power have been associated with increasing difficulty of memory tasks [9].

In addition, Zhang et al. [10] evaluated the modulation effects of mental workload on functional connectivity and found sustained and load-dependent theta power enhancement in the frontal midline region. In addition, Teng et al. [11] compared EEG recorded during resting and active stages in elderly people, and they found that beta spectral power significantly decreased. This change may be a reflection of age-related changes in memory.

1.2. fNIRS as Measurement for Brain Complexity

Near Infrared  Spectroscopy (NIRS) is  a  relatively recent  invention  compared  with EEG. It was developed in 1992 and presently multi-channel systems are commercially available [12]. As a neuroimaging tool, functional NIRS (fNIRS) offers a number of advantages. For example, it is non-invasive, easy to set up and does not require a large space [13]. Moreover, it offers a reasonable trade-off between temporal and spatial resolution [14], improving on the spatial resolution of EEG. This method is also safe for use with neonates [15] and can be applied to observe brain function over extended periods.

To perform fNIRS, two near-infrared lights (650 and 1000 nm) are emitted through the scalp. Photodetectors track the diffusion of NIR light energy through the underlying brain tissue by measuring light intensity. This measurement relies on the absorption of light and light-scattering properties of the tissue. Changes in the metabolic activity of the brain induce haemodynamic responses that alter the rate of NIR light absorption. This influences the level of NIR light intensity measured, which provides an index of oxy-haemoglobin (Hb.O2) and deoxy-haemoglobin (Hb.R) concentrations [16,17]. These are related to local neural function and the presumed cause of metabolic demand.

Functional near-infrared spectroscopy (fNIRS) was used by Borfeld et al. [18] to study brain activity in infants. They found hemispheric asymmetry in the level of Hb.O2, with the left more active while the infants were watching an animation with speech. Schroeter et al. [19] investigated brain functions during a Stroop test using fNIRS. They found a left-lateral difference in the haemodynamic responses of the prefrontal cortex that emerged with age.

Araki et al. [20] used fNIRS to analyse brain function in children diagnosed as hyperactivity disorder in response to treatment with atomoxetine. Levels of Hb.O2 showed a significant decrease in the left ventrolateral prefrontal cortex. This neuroimaging modality has also been applied to investigate changes in brain activity following rehabilitation therapy in stroke patients.

According to Schreppel et al. [21], fNIRS has been widely used to investigate the neural  responses.  They  identified  changes  in  several  brain  regions,  including  prefrontal cortex, occipital region, parietal regions and medial temporal areas. Using fNIRS, it has been found that differences in brain activity of the frontal cortex depend on the level of difficulty of tasks [22]. Additionally, oxy-haemoglobin concentration during difficult tasks changes more when compared with easy tasks. This has been confirmed by research including a study by Sasabe and Hagiwara [23], which characterised and compared oxygenated haemoglobin concentration changes in the brain using fNIRS measurements. Their analysis demonstrated a significant correlation between the number of false answers and the integral of Hb.O2, suggesting that Hb.O2 increases with task difficulty level.

Mirelman et al. [24] also found a difficulty-related increase in frontal brain activity while doing two tasks during walking. Increased Hb.O2 in the frontal cortex was maximal while the participants were walking and serially subtracting 7s followed by walking while counting forward, then the lowest while the participants were walking or standing without performing mental calculations. These kinds of studies rely on well-defined and separable task difficulties; if tasks are ill-defined or too similar, the analysis may be obscured.

1.3. Potential Application in the Study of Memory

Memory plays a crucial role in human life. It allows us to construct life-narratives by linking past, present and future events. Past events that we have experienced are important because they enable us to make educated decisions in the present and to construct mental scenarios in real-time; this forward-thinking, or ability to simulate future events, is a quintessentially human trait [25]. Memory can be classified into long-term, short-term and working memory (WM) [26]. Long-term memory, in which information is held indefinitely, encrypts information semantically for storage. In contrast, short-term memory stores information temporarily for fast response activities. Short-term memory retains information for a short period, whereas WM manipulates it to temporarily store. In other words, WM retains and uses, while short-term merely retains. It has been examined in previous functional neuroimaging studies, which identified the prefrontal cortex (PFC) as the most relevant cortical locus of activation [27]. Memory also changes through time; some grow stronger and last, whereas others fade.

Considering as central to the functioning of the mind, memory is associated with abilities and outcomes such as intelligence and educational achievement and is linked to sensory processes [28]. Different studies have been carried out to understand memory functions based on fNIRS and EEG independently, all of which have their own strengths and weaknesses. For example, Aghajani, Garbey, and Omurtag [29] measured mental workload during a working memory task while recording EEG and fNIRS and identified some clear differences between the two signals. They correctly point out that EEG results from the summed electrical potential of large numbers of cortical and some subcortical neurons with sub-millisecond temporal resolution. In contrast, fNIRS yields localised measures of Hb.O2 and Hb.R concentrations, which are considered to reflect metabolic changes associated with neural activity. Furthermore, the physical measurement of EEG and fNIRS is somewhat different and requires conductive electrodes and light-sensitive transducers, respectively. This distinction in signal acquisition method causes EEG to be prone to distortion from blink and muscle artifacts. However, it is not the case for fNIRS.

Coffey et al. [30] used three-channel fNIRS covering the left forehead together with 8

EEG electrodes placed in the frontal and central areas, in an n-back task. They extracted EEG power spectra and fNIRS Hb amplitude features, and employed them in linear discriminant analysis classification. The results from ten participants revealed a maximum accuracy of

79.7% (with fNIRS) and 89.6% (with EEG). Moreover, Liu et al. [31] employed a 16-optode continuous wave fNIRS system covering the forehead and 28 EEG electrodes positioned according to the standard 10/20 system in the n-back task. The results from sixteen participants showed significant correlations between WM load and EEG as well as Hb.O2 and Hb.R.

The present study focuses on examining brain activity using simultaneous EEG-fNIRS data recorded during a behavioural task. Multiscale entropy (MSE) is then applied to analyse these signals in terms of entropy or brain complexity levels.

Point 2: The literature review is lengthy and cannot point out the key problem, gap, disadvantage or limitation of the current studies that needs to be addressed. The paper does not explain why entropy is needed, why MSE is proposed to analyse the EEG - fNIRS data, and why other neuroimaging modalities are not chosen

Response 2:

Thank you very much for highlighting these. We really appreciate and the manuscript has been improved as follows.

(Lines 151-195)

EEG and fNIRS measure different aspects of brain activity (EEG as   bioelectricity and fNIRS as hemodynamic activity). As mentioned, EEG provides high temporal precision wheareas fNIRS provides spatial resolution. By converting these into a common form (MSE) we aim to combine these signals to share their complementary benefits.

Other neuroimaging modalities (e.g. PET, fMRI, etc.) are less methodologically compatible than EEG and fNIRS in terms of equipment and experimental design.

For this reason, entropy, considered as the complexity of a system, can be used to analysed as a sub or a large system, and the reason to propose MSE for EEG and fNIRS analysis is that MSE encounters a problem in that the statistical reliability of SampEn of a coarse-grained series is reduced as a time scale factor is increased.  If  EEG and fNIRS are adjusted into the same entropy scale, it is possible to employ MSE EEG and MSE fNIRS for brain analysis and therefore create the possibility to merge these two signals into one.

The notion of entropy is first introduced by Rudolph Clausius in the context of thermodynamics [32]. It acts in accordance with the second law of thermodynamics, which states that the change of the entropy in the volume element is equivalent to the ratio of the heat state changes in temperature [33]. The term “Entropy” is thus appreciated from the viewpoint of both thermodynamics and the information theory by means of a display of system uncertainty. The concept of information entropy is proposed by C. E. Shannon, who is regarded as the father of information theory in 1948. At present, the applications of entropy in biomedical signals have received extensive research attentions [34,35,36]. For instance, Pincus [37] proposed the concept of approximate entropy to examine the changes of heart rate for infantile sudden illness. Approximate entropy is good at solving the problem of common signal with short noise in biomedical signals. Richman and Moorman [38]. developed the sample entropy. Sample entropy, compared with approximate entropy, shows better relative consistency and less dependence on data length [38,39,40].

Based on sample entropy with multi-scale approach on temporal data, MSE has been widely applied for medical data analysis. This is pertinent given the complexity and non-linearity of medical data. Shannon entropy has led to the development of various entropies such as Kolmogorov-Sinai entropy, Grassberger entropy and Approximate entropy. Approximate entropy (ApEn) has been commonly used to analyse biological data [41,42]. Furthermore, it has been developed into Sample entropy (SampEn) [38,42]. In the vein of ApEn, SampEn divides the data into short sequences, called epochs, and then determines the sample entropy according to eq. (1).

Equation (1)

where m is template length, r is the tolerance for accepting matches, N is the number of data points, Ai is the number of matches of length m+1, and Bi is number of matches of length m.

To determine MSE, the data are coarse-graining by averaging in the range of a

specified scale; for scale 2, two adjacent data points are averaged, thus the number of total data points is reduced to N/2 if there are originally N data points in total. Through the scaling process, the new data set can be expressed by eq. (2) [38,42,43,45]. Due to a large number of recording channels with a high sampling frequency, multi-scale analysis is required to obtain linear MSE graphs. Sample entropy was therefore selected to optimize efficiency in data processing [38].

Equation (2)

where τ is the scale level, and xis the original data point then, the new data set is used to compute sample entropy in each scale.

Previous studies have shown that SampEn has the acceptable range for m= 2, r= 0.15 x SD of the time series [38,46]. Adopting the method proposed by Catarino et al. [46], we used m = 2, r = 0.15 x SD of the time series. For N, we calculated from the sampling frequency  and  epoch  length  used  for  quantitative  analysis  (12  seconds).  According  to Richman and Moorman [38], the appropriate N size for SampEn analysis must be a minimum of 10to 20m, where m = 2; N should be at least 100 to 400 in each scale. When the scale increases, the number of N in each scale will decrease. For EEG sampling frequency at 500

Hz, when calculating, the size of N, NEEG  is equal to 500 Hz x 12 sec = 6000 data points. Considering the 20th scale, the size of the NEEG is 300, which is still enough for SampEn analysis. With respect to fNIRS, which has a sampling frequency of only 10 Hz. To calculate the size of N, NfNIRS  = 10 Hz x 12 sec = 60 data points, which is not enough to perform analysis with SampEn from Scale 1. Therefore, it is necessary to up-sample by a factor of 10 using the spline interpolation method to get NfNIRS  = 1200 data points, which is enough to

analyse with SampEn up to Scale 12. For topographic rendering, the values of MSE EEG and MSE fNIRS at Scale 10 were used for analysis as the mean values with the sufficient size of N.

Section 2: Multiscale Entropy

Point 3: The abbreviation MSE has been used for multiscale entropy, it is now used for multiscale sample entropy although multiscale sample entropy and multiscale entropy are not the same.

Response 3:

Multiscale sample entropy has been replaced by multiscale entropy. Thank you for spotting this detail.

Point 4: Different entropy functions in the literature are reviewed in a very short paragraph and then Sample Entropy is introduced for multiscale entropy. It is not clear why those entropy functions cannot be applied, and why only Sample Entropy is chosen.

Response 4:

Thank you very much for your expertise. The manuscript has been improved based on your suggestion.

(Lines 173 – 178)

To determine MSE, the data are coarse-graining by averaging in the range of a specified scale; for scale 2, two adjacent data points are averaged, thus the number of total data points is reduced to N/2 if there are originally N data points in total. Through the scaling process, the new data set can be expressed by eq. (2) [38,42,43,45]. Due to a large number of recording channels with a high sampling frequency, multi-scale analysis is required to obtain linear MSE graphs. Sample entropy was therefore selected to optimize efficiency in data processing [38].

Point 5: A theoretical background on entropy and a comparison between entropy functions are required.

Response 5:

Thank you very much for your constructive suggestion. A theoretical background on entropy and a comparison between entropy functions have been included as follows.

(Lines 151 -162)

The notion of entropy is first introduced by Rudolph Clausius in the context of thermodynamics [32]. It acts in accordance with the second law of thermodynamics, which states that the change of the entropy in the volume element is equivalent to the ratio of the heat state changes in temperature [33]. The term “Entropy” is thus appreciated from the viewpoint of both thermodynamics and the information theory by means of a display of system uncertainty. The concept of information entropy is proposed by C. E. Shannon, who is regarded as the father of information theory in 1948. At present, the applications of entropy in biomedical signals have received extensive research attentions [34,35,36]. For instance, Pincus [37] proposed the concept of approximate entropy to examine the changes of heart rate for infantile sudden illness. Approximate entropy is good at solving the problem of common  signal  with  short  noise  in  biomedical  signals.  Richman  and  Moorman  [38]. developed the sample entropy. Sample entropy, compared with approximate entropy, shows better relative consistency and less dependence on data length [38,39,40].

Point 6: The sentence on lines 154 and 155 “where m is template length, r is the tolerance for accepting matches, N is the number of data points, Ai is the number of matches of length m+1, and Bi is number of matches of length m” needs to be revised (remove indentation, correct subscript for Ai and Bi, and use italic font for m, N, Ai, and Bi). The same for the sentence on lines 160 and 161.

Response 6:

The sentences on lines 154, 155, 160 and 161 have been corrected. Thank you very much for your kind suggestion.

Note: (Lines 154-155) have been changed to (Lines 171-172).

where m is template length, r is the tolerance for accepting matches, N is the number of data points, Ai is the number of matches of length m+1, and Bi is number of matches of length m.

Note: (Lines 160-161) have been changed to (Lines 180-195).

where τ is the scale level, and xis the original data point then, the new data set is used to compute sample entropy in each scale.

Previous studies have shown that SampEn has the acceptable range for m= 2, r= 0.15 x SD of the time series [38,46]. Adopting the method proposed by Catarino et al. [46], we used m = 2, r = 0.15 x SD of the time series. For N, we calculated from the sampling frequency and epoch length used for quantitative analysis (12 seconds). According to Richman and Moorman [38], the appropriate N size for SampEn analysis must be a minimum of 10m  to

20m, where m = 2; N should be at least 100 to 400 in each scale. When the scale increases, the number of N in each scale will decrease. For EEG sampling frequency at 500 Hz, when

calculating, the size of N, NEEG is equal to 500 Hz x 12 sec = 6000 data points. Considering the 20th  scale, the size of the NEEG  is 300, which is still enough for SampEn analysis. With respect to fNIRS, which has a sampling frequency of only 10 Hz. To calculate the size of N, NfNIRS = 10 Hz x 12 sec = 60 data points, which is not enough to perform analysis with SampEn from Scale 1. Therefore, it is necessary to up-sample by a factor of 10 using the spline interpolation method to get NfNIRS = 1200 data points, which is enough to analyse with SampEn up to Scale 12. For topographic rendering, the values of MSE EEG and MSE fNIRS

at Scale 10 were used for analysis as the mean values with the sufficient size of N.

Section 3: Materials and Methods.

Point 7: The research design is good. The number of participants is small.

Response 7:

Thank you very much for your kindness and support.

Sample size in this study is small (less than 30) so that an assumption of normal distribution is impossible to be met. For this case, the use of the Wilcoxon signed-rank test, which is a substitution  of  paired  sample  t-test  in  parametric  statistics,  is  therefore  considered  as  a suitable method of data analysis.

Point 8: It would be helpful for other researchers to use the collected data set if it is made publicly available.

Response 8:

Thank you very much for your kindness and support. We totally agree that it would be helpful for other researchers to use the collected data set if it is made publicly available.

Point 9: What were the EEG and fNIRS sampling frequencies?

Response 9:

Thank you very much for highlighting these. The manuscript has been improved as follows. (Lines 252-258)

EEG data was recorded using a Neurofax 1100 (Nihonkoden, Japan) at a sampling frequency of 500 Hz. All recorded EEG channels were arranged into a topographic figure. Independent component analysis (ICA) was performed on raw EEG-fNIRS data using the open-source toolbox EEGLAB (v12.0.2.5b [50,51] in MATLAB (R2015b; Mathworks Ltd.; US). Components relating to eye blinking and eye movement artifacts were removed. Other noises were also removed with digital bandpass (0.5 to 50 Hz) and notch (60 Hz) filters. Pre- stimulus baseline correction was then performed.

All fNIRS data were recorded by fNIRS Hitachi ETG-7100 (Hitachi, Japan) with 10 Hz sampling frequency.

Section 4: Results

Point 10: The multiscale entropy was already used as seen in references 35-38. However, there are no results for comparison of the proposed multiscale entropy in this paper with the current one in those references.

Response 10:

Thank you very much for your kind suggestion. We perform an analysis of EEG and NIRS data using multiscale entropy as proposed in Costa et al. (2002) cited in Reference 45. We adopted the same method of analysis using a different experimental design.

Section 5: Discussion and Conclusions.

Point 11: The research findings and discussions are interesting.

Response 11:

Thank you very much for your kindness and support. We really appreciate and feel honoured.

This study was not all plain sailing as no part came smoothly, effortlessly and painlessly. Some of the challenges were managerial issues, while others were research-based in nature. Nonetheless, there were some significant and exciting findings emerged from this study. We believe it is worth sharing so that other researchers could benefit from this study.

.........................End of response to reviewer................................................

Dear Reviewer2,

First of all, we would like to thank you for your time and kind suggestion. We really appreciate and feel honoured. This study is not all plain sailing as no part came smoothly, effortlessly and painlessly. Some of the challenges were managerial issues, while others were research-based in nature. Nonetheless, there were some significant and exciting findings emerged from this study. We believe it is worth sharing so that other researchers could benefit from this study. We truly hope this manuscript will be helpful for other researchers if it is made publicly available.

Secondly, we have improved the manuscript based on your suggestion and comments. However, due to the major revision, this manuscript has totally been adjusted and improved. For this reason, we could not provide track changes or highlights as expected because this revision has been adjusted to a totally new, improved one. We have tried to use track changes for response to reviewers, but it is very difficult to read, spot and understand.  Therefore, in response to reviewers, we have provided all (including lines and sentences) in the form of “Point and Response to reviewers” separately into each point to improve and hopefully it will easily be read and spotted. Please find the attachment of “Responses to Reviewers”.

Finally, it would be really appreciated if you could provide the verification before 4th January. My name is Thanate Angsuwatanakul, the first author of this manuscript. I am currently a PhD student, Kyushu University, and now in the process of VIVA examination. The date of VIVA examination is on 6th January 2020 at 3.30 p.m. However, I am still not qualified to attend the exam until I submit the proof that my manuscript is accepted for publication in reputable refereed journals. The main problem is my candidature term end is 10th January 2020 meaning that I have about one week for the process of VIVA examination. I realize that this situation is very difficult not only for me but for everyone including the reviewers for kind assistance to verify this manuscript. Honestly, I realize that this manuscript may be rejected from the reviewers meaning that I will be terminated very soon. However, I will not regret because my intention is I would like to submit (publish) in “Entropy” once in my life and I have already done my best. 

Thank you very much for your kind assistance and consideration. I look forward to hearing from you soon.

Sincerely yours,

Thanate Angsuwatanakul

Round 2

Reviewer 2 Report

The revised paper is very good and ready for publication. Thank you for the revision.